# Holistic Unlearning Benchmark: A Multi-Faceted Evaluation for Text-to-Image Diffusion Model Unlearning

## Abstract

As text-to-image diffusion models become advanced enough for commercial applications, there is also increasing concern about their potential for malicious and harmful use. Model unlearning has been proposed to mitigate the concerns by removing undesired and potentially harmful information from the pre-trained model. So far, the success of unlearning is mainly measured by whether the unlearned model can generate a target concept while maintaining image quality. However, unlearning is typically tested under limited scenarios, and the side effects of unlearning have barely been studied in the current literature. In this work, we thoroughly analyze unlearning under various scenarios with five key aspects. Our investigation reveals that every method has side effects or limitations, especially in more complex and realistic situations. By releasing our comprehensive evaluation framework with the source codes and artifacts, we hope to inspire further research in this area, leading to more reliable and effective unlearning methods.

## 1 Introduction

Text-to-image diffusion models have achieved remarkable success in various real-world applications based on the huge number of text-to-image pairs used for training (Ramesh et al., 2022; Dhariwal & Nichol, 2021; Nichol et al., 2021; Podell et al., 2023). Since these pairs are often sourced from the Internet, they may include violent, harmful, or unethical content (Schuhmann et al., 2022). Previous studies show that large-scale text-to-image models can generate malicious images (Rando et al., 2022; Ma et al., 2024a; Kim et al., 2024; Yang et al., 2024b). Most models use a safety filter to block the generation of undesirable content (Saharia et al., 2022; Rombach et al., 2022; Ramesh et al., 2022). However, the safety filters can only work with a predefined set of malicious patterns and may be vulnerable to generated images with unseen patterns.

As a practical solution, text-to-image unlearning methods have been proposed (Gandikota et al., 2023; 2024; Kumari et al., 2023; Heng & Soh, 2024; Fan et al., 2023; Huang et al., 2024). These methods aim to modify pre-trained models to prevent the generation of images containing specific target concepts. While promising, the evaluation of previous methods has been limited to verifying the absence of target concepts and assessing the visual quality of generated images. This narrow focus often overlooks other critical factors, such as unintended side effects or performance degradation on unrelated concepts. Without a comprehensive evaluation framework, it remains challenging to assess and compare the ability of unlearning methods, leaving important questions about their effectiveness and limitations unanswered.

To address these limitations, we propose *Holistic Unlearning Benchmark (HUB)* that systematically assesses unlearning methods across five key aspects, as summarized in Table 1: *effectiveness on target concepts, faithfulness of images, compliance with prompts, robustness on side effects*, and *consistency in downstream applications*. By examining each of these dimensions, HUB provides an in-depth assessment from multiple perspectives, highlighting both their strengths and weaknesses.

In this paper, we evaluate six state-of-the-art methods, including ESD (Gandikota et al., 2023), UCE (Gandikota et al., 2024), AC (Kumari et al., 2023), SA (Heng & Soh, 2024), SalUn (Fan et al., 2023), and Receler (Huang et al., 2024). Our empirical experiments reveal that *no method works well in all aspects of evaluations*, underscoring the need for more holistic unlearning approaches. By

| Perspective | Task | Section | AC | SA | SalUn | UCE | ESD | Receler | Ours |
|---|---|---|---|---|---|---|---|---|---|
| Effectiveness on target concept | Simple prompt | Section 4.1 | ○ | ○ | ○ | ○ | ○ | ○ | ○ |
| | Diverse prompt | Section 4.1 | △ | ○ | | | | ○ | ○ |
| Faithfulness of images | Simple prompt | Section 4.2 | | | | | | | ○ |
| | Diverse prompt | Section 4.2 | | | | | | | ○ |
| | MS-COCO prompt | Section 4.2 | ○ | ○ | ○ | ○ | ○ | ○ | ○ |
| Compliance with prompt | MS-COCO alignment | Section 4.2 | ○ | | | ○ | ○ | ○ | ○ |
| | Selective alignment | Section 4.3 | | | | | | | ○ |
| Robustness on side effects | Over-erasing effect | Section 5.1 | | | | | △ | | ○ |
| | Model bias | Section 5.2 | | | | | | | ○ |
| Consistency in downstream applications | Sketch-to-image | Section 6.1 | | | | | | | ○ |
| | Image-to-image | Section 6.1 | | | | | | | ○ |
| | Concept restoration | Section 6.2 | | | | | | | ○ |

Table 1: Comparison of different evaluation tasks in previous studies and this work. Our benchmark covers a wide range of tasks. △ indicates that the method is evaluated qualitatively only without any quantitative metric.

providing this comprehensive evaluation framework, we not only shed light on the current limitations but also pave the way for future research to develop more effective and reliable unlearning methods. Additionally, we will release the evaluation code and datasets to support further exploration in this area, hoping to inspire new research directions.

## 2 RELATED WORK

There is a growing body of research on unlearning techniques for pre-trained text-to-image models, aiming to mitigate the generalization of inappropriate images. ESD (Gandikota et al., 2023) propose a fine-tuning method that inversely guides the model against generating a specified target concept text. AC and SalUn (Kumari et al., 2023; Fan et al., 2023) proposes a fine-tuning method that can map the target concept to alternative concepts. UCE (Gandikota et al., 2024) fine-tunes cross-attention layers for unlearning. SA (Heng & Soh, 2024) introduces an unlearning based on continual learning, and Receler (Huang et al., 2024) uses an adapter and a masking scheme.

Several methods have been proposed to evaluate the robustness of text-to-image diffusion models against malicious uses. Most of these approaches focus on optimizing soft prompts to generate undesired concepts (Pham et al., 2023; Ma et al., 2024a; Tsai et al., 2023; Zhang et al., 2024b; Chin et al., 2024; Yang et al., 2024a; Rando et al., 2022; Yang et al., 2024b), or directly finding prompts that lead to unwanted content generation (Kim et al., 2024). Although these methods assess model vulnerabilities, they have two main limitations: 1) they only provide a prompt engineering view of model robustness, and 2) they require access to the model weights, which may not always be possible. In contrast, our proposed benchmark offers a more comprehensive evaluation from multiple perspectives without requiring access to the model weights.

Furthermore, some benchmarks have been introduced to evaluate unlearned models (Ma et al., 2024b; Zhang et al., 2024a; Schramowski et al., 2023). For example, Schramowski et al. (2023) introduce the I2P dataset to assess the ability of a model to avoid generating inappropriate content that could be offensive, insulting, or anxiety-inducing. Zhang et al. (2024a) introduce a stylized image dataset to evaluate models that have undergone style unlearning. Similarly, Ma et al. (2024b) offer a copyright dataset to measure how effectively an unlearned model protects copyrighted material by not reproducing protected content. While previous benchmarks have primarily focused on providing datasets to evaluate models, our benchmark goes beyond this by addressing unintended changes in model behavior and potential problems that can occur when unlearned models are applied in real-world applications.

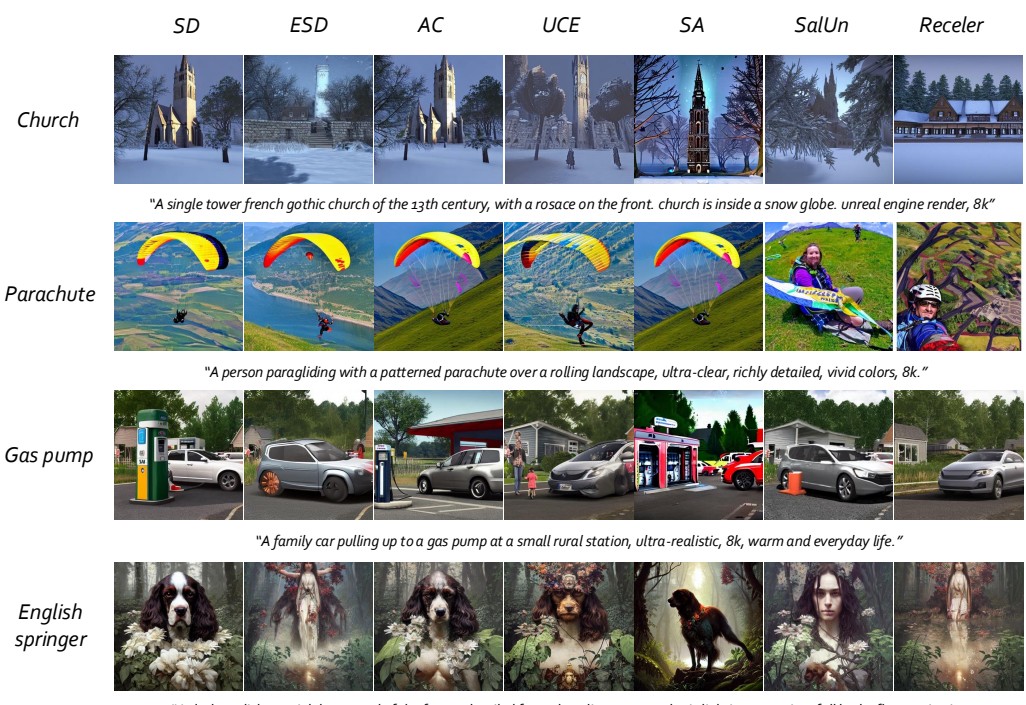

| SD | ESD | AC | UCE | SA | SalUn | Receler |

*Church*

"A single tower french gothic church of the 13th century, with a rosace on the front. church is inside a snow globe. unreal engine render, 8k"

*Parachute*

"A person paragliding with a patterned parachute over a rolling landscape, ultra-clear, richly detailed, vivid colors, 8k."

*Gas pump*

"A family car pulling up to a gas pump at a small rural station, ultra-realistic, 8k, warm and everyday life."

*English springer*

"A dark english spaniel dog as god of the forest, detailed face, clean lines, atmospheric lighting, amazing, full body, flowers, intricate, highly detailed, digital painting, artstation, concept art, sharp focus, illustration, art by greg rutkowski and alphonse mucha"

Figure 1: Failure samples of generated images from diverse prompts containing each target concept.

## 3 EXPERIMENTAL SETTING

**Concept unlearning.** In this work, we focus on *concept* unlearning, where the concept is defined as a single English word or phrase. We refer to the concept to be unlearned as the *target concept*. Following previous works (Gandikota et al., 2023; 2024; Fan et al., 2023), we select four target concepts from the Imagenette dataset (Howard, 2019): church, parachute, gas pump, and English springer covering diverse and exhaustive scenarios. These objects also vary in conceptual complexity—from broad categories to specific entities—allowing us to analyze unlearning performance across different levels of specificity.

**Baseline.** We use Stable Diffusion 1.4 (Rombach et al., 2022) as our baseline model and apply six unlearning methods: AC (Kumari et al., 2023), SA (Heng & Soh, 2024), SalUn (Fan et al., 2023), UCE (Gandikota et al., 2024), ESD (Gandikota et al., 2023), and Receler (Huang et al., 2024). Based on their approach to handling the target concept, these methods are categorized into 1) mapping-based approaches, which replace the target concept with an *alternative concept* during unlearning (AC, SA, SalUn), and 2) non-mapping-based approaches, which utilize techniques like gradient ascent on the target concept (UCE, ESD, Receler). For AC and SA, we select bird as the alternative concept since it has easily distinguishable and distinct features. Detailed explanations and configurations for each method are provided in Appendix A and Appendix B, respectively.

## 4 REVISIT CURRENT EVALUATION OF TEXT-TO-IMAGE UNLEARNING

In text-to-image diffusion model unlearning, evaluations have usually focused on two aspects: 1) whether the model successfully avoids generating a target concept and 2) whether the visual quality measure, such as FID, is maintained. In this section, we expand the previous evaluation methods with additional and newly developed tasks that can measure the performance from various perspectives.

| | Church | | | Parachute | | | Gas pump | | | English springer | | |
|---|---|---|---|---|---|---|---|---|---|---|---|---|
| | Simple | Diverse | Δ | Simple | Diverse | Δ | Simple | Diverse | Δ | Simple | Diverse | Δ |
| Original | 1.000 | 0.967 | -0.033 | 0.992 | 0.953 | -0.039 | 0.944 | 0.920 | -0.024 | 0.999 | 0.982 | -0.017 |
| AC | 0.110 | 0.905 | 0.795 | 0.003 | 0.562 | 0.559 | 0.000 | 0.663 | 0.663 | 0.000 | 0.034 | 0.034 |
| SA | 0.333 | 0.792 | 0.459 | 0.108 | 0.787 | 0.679 | 0.000 | 0.337 | 0.337 | 0.000 | 0.031 | 0.031 |
| SalUn | 0.000 | 0.132 | 0.132 | 0.000 | 0.124 | 0.124 | 0.000 | 0.030 | 0.030 | 0.000 | 0.040 | 0.040 |
| UCE | 0.146 | 0.469 | 0.323 | 0.014 | 0.109 | 0.095 | 0.000 | 0.030 | 0.030 | 0.007 | 0.031 | 0.024 |
| ESD | 0.032 | 0.348 | 0.316 | 0.017 | 0.202 | 0.185 | 0.000 | 0.043 | 0.043 | 0.003 | 0.032 | 0.029 |
| Receler | 0.000 | 0.045 | 0.045 | 0.001 | 0.012 | 0.011 | 0.000 | 0.010 | 0.010 | 0.001 | 0.002 | 0.002 |

Table 2: Proportion ($\downarrow$) of generated images that include the target concept. Simple and diverse prompts are used for the generation. $\Delta$ refers to the difference in target proportion between simple and diverse prompts.

## 4.1 How robust is the unlearning under diverse and complex prompts?

Previous studies have evaluated unlearning methods with images generated from prompts generated via template "*a photo of a* {concept}"[1], where {concept} is replaced by a target (Gandikota et al., 2023; 2024; Heng & Soh, 2024; Fan et al., 2023). However, in real-world scenarios, users often generate images with detailed and complex prompts that may generate the target concept indirectly. Therefore, even if the unlearning performs well with straightforward prompts, it may fail when faced with more diverse and realistic prompts.

To address this, we curate 100 prompts for each concept by generating diverse prompts from GPT-4o, incorporating non-English instructions. Given an initial prompt set, we manually inspect prompts to curate the final 100 prompts, denoted as 'diverse prompts' in experiments. While prior studies have utilized LLM-based paraphrasing (Kumari et al., 2023; Lu et al., 2024), our approach goes further by creating more sophisticated and complex prompts. The detailed GPT-4o instruction, curation process, and examples of curated prompts are provided in Appendix C.2.

We generate ten images per diverse prompt with different random seeds, resulting in 1,000 generated images for each target concept. We also generate 1,000 images with simple prompts. We measure the proportion of images containing the target concept using GPT-4o, which has demonstrated high accuracy in identifying visual objects (Wu et al., 2023; 2024). The instruction to classify the target concept is provided in Appendix C.1.

Table 2 shows the proportion of the generated images with the four target concepts across six unlearning methods. Among the methods, Receler and SalUn demonstrate the most robust performance, maintaining low target concept generation regardless of prompt diversity. In contrast, methods such as AC and SA reveal significant differences between simple and diverse prompts, indicating weaker generalization to more complex and realistic prompts.

We observe that the specificity of the target concept is strongly related to the success of unlearning in terms of target proportion. Broad concepts like church or parachute tend to reappear with diverse prompts, making them harder to erase. In contrast, more specific concepts such as English springer, which lack synonyms or related terms, do not consistently appear with simple and diverse prompts. We also present the qualitative results in Fig. 1, showing that the target concept is generated from diverse prompts. More samples can be found in Appendix C.2.

## 4.2 How does unlearning change the quality of generated images?

To evaluate the quality of the generated images from unlearned models, we generate and categorize images into two:

- A set of natural images generated with the captions of MS-COCO 30K dataset (Lin et al., 2014) as prompt. We assess the image quality from two perspectives: visual quality of the images and alignment between images and prompts. We measure the visual quality of the generated images using FID. The alignment score between the generated images and corre-

---

[1]We call the prompt generated by this template a 'simple prompt'

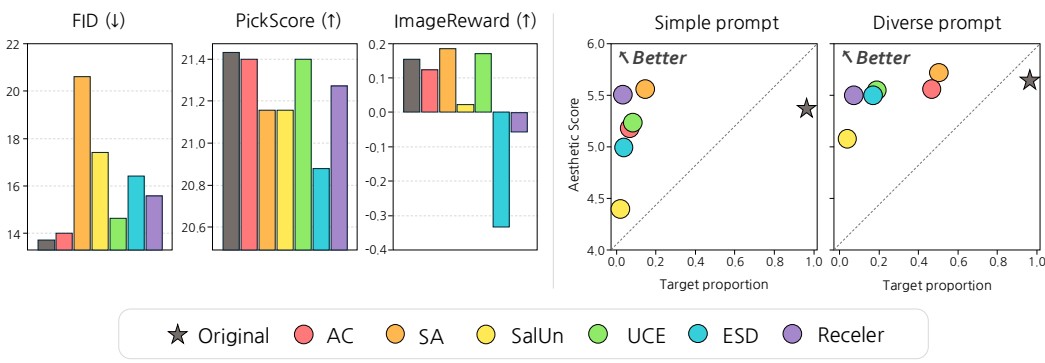

Figure 2: (Left) visual image quality (FID) and alignment scores (PickScore and ImageReward) of images generated from the captions of MS-COCO 30K dataset. (Right) Target proportion vs. image quality of the target-related images generated from simple and diverse prompts.

| | Original | AC | SA | SalUn | UCE | ESD | Receler |
|---|---|---|---|---|---|---|---|
| Church | 0.961 | 0.953 (-0.008) | 0.968 (+0.007) | 0.895 (-0.066) | 0.973 (+0.012) | 0.921 (-0.040) | 0.929 (-0.032) |
| Parachute | 0.938 | 0.871 (-0.067) | 0.789 (-0.149) | 0.712 (-0.226) | 0.910 (-0.028) | 0.939 (+0.001) | 0.939 (+0.001) |
| Gas pump | 0.886 | 0.890 (+0.004) | 0.756 (-0.130) | 0.697 (-0.189) | 0.946 (+0.060) | 0.926 (+0.040) | 0.975 (+0.089) |
| English springer | 0.909 | 0.788 (-0.121) | 0.653 (-0.256) | 0.752 (-0.157) | 0.937 (+0.028) | 0.836 (-0.073) | 0.897 (-0.012) |

Table 3: Proportion of generated images containing the correct background. The number in parentheses indicates the difference in proportion compared to the original model.

sponding captions is evaluated using PickScore (Kirstain et al., 2023) and ImageReward (Xu et al., 2024).

- A set of images generated from the simple and diverse prompts used in Section 4.1. We assess the quality of these images using an aesthetic score (Schuhmann et al., 2022).

In Fig. 2(left), we report the FID, PickScore, and ImageReward for the natural images. We observe the FID score decrease across all methods compared to the `Original`. Especially, `SA` and `SalUn` show relatively low image quality. The visual quality of generated images is not always correlated with the alignment scores. For example, `SA` performs the worst in FID but the best in ImageReward. `ESD` performs relatively well in FID but shows the worst performance in alignment scores. These results suggest that FID, widely used in previous studies, should not be the only metric for image quality evaluation.

In Fig. 2(right), we plot the aesthetic scores along with the target proportion for the target-related images. The results show the trade-off between the target proportion and the image quality. For instance, while `SalUn` effectively removes the target concept, the image quality is noticeably lower than the other methods.

## 4.3 CAN UNLEARNING SELECTIVELY REMOVE THE TARGET CONCEPT FROM A PROMPT?

While we have focused on evaluating visual quality and alignment, one critical aspect has not yet been addressed: how the model handles prompts containing the target concept. It is essential to evaluate whether the model can selectively remove the target concept while accurately generating the rest of the prompt. To address this, we introduce the *selective alignment task*, which assesses the ability of models not to generate a target concept while retaining the other components of the prompt. For example, given prompt "church *with a* beach", the model should ignore the church but still generate a correct image of the beach.

For this task, we generate images with prompts from template "{concept} *with a* {background}" with varying target concepts and backgrounds. We curate ten backgrounds listed in Table 10. We use GPT-4o to assess whether the backgrounds are correctly generated. As shown in Table 3, `UCE`

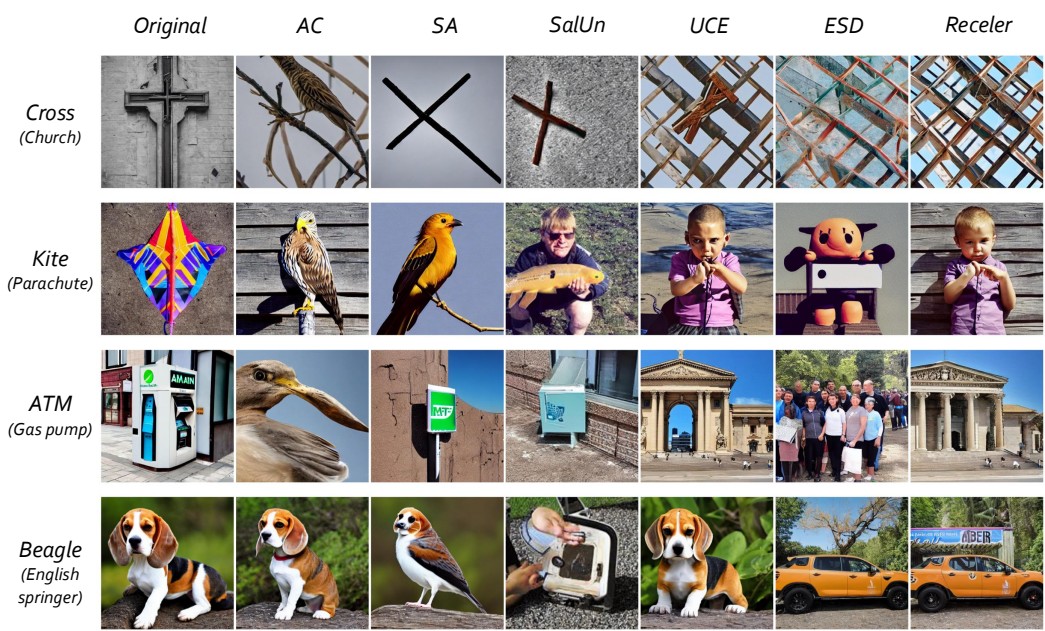

Figure 3: Over-erasing effect. When the target concepts are church, parachute, gas pump and English springer, related concepts such as cross, kite, ATM and beagle are also erased from some unlearned models, respectively.

and `Receler` generally perform the best, consistently maintaining the background scene. In contrast, `SalUn` and `SA` show a drop in performance.

> 💡 **Takeaway.** To comprehensively evaluate the performance of the unlearned model, it is necessary to 1) assess its effectiveness using diverse and complex prompts, and 2) extend the evaluation beyond the faithfulness of images to also consider how well the model complies with the entire prompt.

Based on the findings from this section, we observe that `Receler` and UCE perform more robustly than the others. However, a key question remains. *Is there any side effect of the excellent robustness?* We investigate further to address the unknown side effects in the following section.

## 5 SIDE EFFECTS OF UNLEARNING

We extend our analysis to examine how unlearning affects the 1) generation of related concepts, such as the generation of a cross in a church-erased model, and 2) the underlying population distribution estimated by the generative model, *e.g.*, the distribution of samples without any text condition.

### 5.1 WHAT IS THE INFLUENCE OF UNLEARNING ON RELATED CONCEPTS?

Existing unlearning methods evaluate whether unrelated concepts remain unaffected after the target concept is erased, such as generating gas pump images from a church-erased model (Gandikota et al., 2023; 2024; Fan et al., 2023; Huang et al., 2024). However, this evaluation often overlooks the impact on semantically or visually similar concepts. Unlearning a target concept can lead to unintended consequences on related concepts, a phenomenon we refer to as the *over-erasing effect*. This occurs due to shared feature representations inherent in such models (Radford et al., 2021). For instance, removing the concept church may also degrade the generation of target-related concepts like cross, altar, or bible, as they share latent features with church. Addressing this issue is essential, as it limits the capacity of models to generate accurate and diverse content after unlearning, posing a significant challenge for future research.

|  | Original | AC | SA | SalUn | UCE | ESD | Receler |
|---|---|---|---|---|---|---|---|
| Church | 0.965 | 0.806 (-0.159) | 0.697 (-0.268) | 0.707 (-0.258) | 0.903 (-0.062) | 0.451 (-0.514) | 0.607 (-0.358) |
| Parachute | 0.951 | 0.763 (-0.188) | 0.833 (-0.118) | 0.623 (-0.328) | 0.899 (-0.052) | 0.514 (-0.437) | 0.508 (-0.443) |
| Gas pump | 0.959 | 0.811 (-0.148) | 0.672 (-0.287) | 0.245 (-0.714) | 0.667 (-0.292) | 0.151 (-0.808) | 0.087 (-0.872) |
| English springer | 0.998 | 0.989 (-0.009) | 0.659 (-0.339) | 0.460 (-0.538) | 0.912 (-0.086) | 0.371 (-0.627) | 0.522 (-0.476) |

Table 4: Proportion of generated images containing target-related concepts. The number in parentheses indicates the difference in proportion compared to the original model.

To assess the over-erasing effect, we curate five related concepts for each target concept and generate 100 images for each concept. We measure the proportion of related concepts in the generated images using GPT-4o. The full list of related concepts for each target is provided in Appendix E. As shown in Table 4, all unlearning methods lead to a decrease in related concept generation compared to the original model. Notably, Receler and ESD have the most significant negative impact, while AC and UCE show smaller reductions. While Receler and ESD effectively remove the target concept as shown in Section 4.1, they often introduce unintended effects on related concepts.

Fig. 3 shows samples of generated images from prompts containing target-related concepts. Mapping-based methods like AC, SA, and SalUn often generate alternative concepts, such as a bird instead of a beagle, or a golfball in place of an ATM. Additionally, Receler and ESD demonstrate severe distortions, further highlighting the over-erasing effect by producing irrelevant content for related concepts. Additional qualitative results can be found in Appendix E.

> 💡 **Takeaway.** Unlearning removes the target concept together with target-related concepts. Hence, the over-erasing effect can limit the diversity of the model outputs on target-related concepts.

|  | Unconditional | | | Conditional |
|---|---|---|---|---|
|  | $\mathcal{D}_{KL}$ ($\downarrow$) | Target ($\downarrow$) | Alternative ($\downarrow$) | Target ($\downarrow$) |
| ESD | 0.070 | 0.017 | ✗ | 0.030 |
| SA | 0.014 | 0.091 | 0.000 | 0.000 |
| SalUn | 0.374 | 0.009 | 0.417 | 0.003 |
| AC | 2.023 | 0.002 | 0.877 | 0.003 |

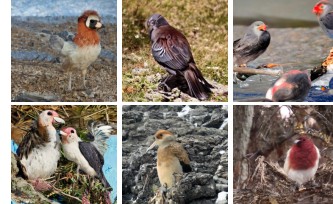

Table 5: Average KL-divergence between the original and unlearned models for four unlearning methods and proportions of generated images containing target and alternative classes in the conditional and unconditional generations.

Figure 4: Images from AC and SA unconditionally. The target is church, and the alternative concept is bird.

## 5.2 HOW DOES UNLEARNING CHANGE THE UNDERLYING ESTIMATED DISTRIBUTION?

The over-erasing effect raises a natural question about the changes in the estimated density of the generative models, as the primary goal of a generative model is to estimate the true data distribution from the training set. This change could lead to a shift in the overall distribution, affecting its ability to generate images. To answer this, we investigate how unlearning changes the estimated density of the generative models.

**Experimental settings.** We use a diffusion model trained on the MNIST dataset rather than natural images since the effects of unlearning are well pronounced with a limited set of classes. We adopt the joint training method from classifier-free guidance (Ho & Salimans, 2022), enabling conditional and unconditional generation. For each unlearning method, we train ten models, each unlearning one of the classes from zero to nine. We exclude UCE and Receler from this experiment since they rely on cross-attention layers, which are absent in the DDPM architecture. Detailed training procedures for each unlearning method are provided in Appendix B.2.

To compare the underlying density differences between the original and unlearned models, we analyze the class distributions of images generated unconditionally, as it is impossible to measure the density from a model directly. We 1) unconditionally sample 10,000 images from each model, 2) classify them, and 3) compute the KL divergence of non-target class distributions between original and

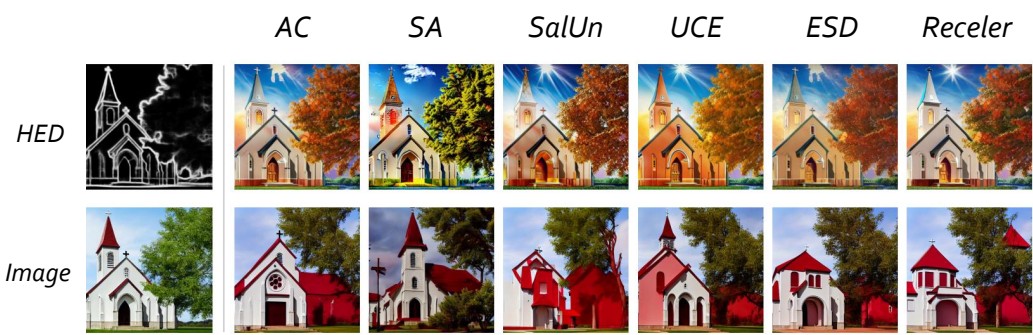

Figure 5: Generated samples with image conditions shown in the leftmost column. HED and reference image are used as visual conditions of the generation. More samples can be found in Appendix G.2.

unlearned models. We use a LeNet-5 classifier (LeCun et al., 1998) with 98% test accuracy. We also report the target and alternative class proportions in the generated images. Unlike in Stable Diffusion, which maps the target class to a different class, SA maps it to a uniform distribution for this experiment, followed by the original implementation. As a comparison, we report the average proportion of the target class in the images of class-conditional generation.

**Results.** Table 5 displays the average KL divergence and class proportions in the generated images. The proportion of the alternative class in unconditional generation reflects this distortion. For the case of SalUn and AC, roughly 41% and 87% of unconditionally generated images belong to the alternative class, indicating a bias towards it. From the KL divergence view, the distributions obtained from AC and SalUn are the most distorted from the original distribution. In contrast, the distributions of ESD and SA are changed less after unlearning than those of SalUn or AC.

An interesting observation is that the unconditional distribution of SA remains almost unchanged from the original model shown by the KL divergence and target proportion, meaning that SA still generates the target concept unconditionally. The result implies that the SA method is tightly connected with the input condition and only alters the conditional distribution.

In addition, Fig. 4 shows the images generated unconditionally from Stable Diffusion after unlearning with AC and SA. Unlearned models with mapping-based methods generate the alternative concept more frequently than the original model. We also observe that SA is biased when applied to Stable Diffusion. In this case, similar to other mapping-based methods, SA maps the target concept to another concept rather than a uniform noise. Since the bias in generative models can impact their capabilities and lead to unexpected behavior, the problem needs to be addressed in future work. Additional explanations and image samples are provided in Appendix F.1.

> 💡 **Takeaway.** Existing unlearning methods alter the underlying distribution. SA minimizes bias in MNIST, but it still leads to bias with more complex distributions such as Stable Diffusion.

## 6 EFFECT OF UNLEARNING TO DOWNSTREAM TASKS

### 6.1 DO UNLEARNED MODELS WORK CONSISTENTLY IN DOWNSTREAM TASKS?

Text-to-image diffusion models are widely used in *downstream tasks* such as image editing and customization, often with additional conditions, such as reference images or semantic layouts. While most existing research on unlearning has focused on scenarios where only text-based prompts are provided, it is important to ensure that unlearning is equally effective for downstream tasks with additional conditions. To verify this, we conduct evaluations on sketch-to-image and image-to-image tasks.

**Experimental settings.** We employ ControlNet (Zhang et al., 2023) with HED (Xie & Tu, 2015) and ControlNet reference-only (Zhang et al., 2023) to mimic the sketch-to-image and image-to-image

|  | Church | Parachute | Gas pump | English springer | Average |  | Church | Parachute | Gas pump | English springer | Average |
|---|---|---|---|---|---|---|---|---|---|---|---|
| AC | 1.000 | 0.884 | 0.977 | 0.137 | 0.750 | AC | 0.997 | 0.984 | 0.632 | 0.073 | 0.672 |
| SA | 0.991 | 0.961 | 0.864 | 0.044 | 0.715 | SA | 0.987 | 0.755 | 0.085 | 0.025 | 0.463 |
| SalUn | 0.948 | 0.405 | 0.944 | 0.429 | 0.682 | SalUn | 0.495 | 0.747 | 0.404 | 0.251 | 0.474 |
| UCE | 1.000 | 0.873 | 0.963 | 0.792 | 0.907 | UCE | 0.977 | 0.989 | 0.655 | 0.868 | 0.872 |
| ESD | 1.000 | 0.855 | 0.913 | 0.697 | 0.866 | ESD | 0.963 | 0.969 | 0.565 | 0.795 | 0.823 |
| Receler | 0.992 | 0.853 | 0.469 | 0.400 | 0.679 | Receler | 0.793 | 0.964 | 0.111 | 0.372 | 0.560 |

Table 6: Proportion of generated images containing target concept generation when a sketch (left) and a reference image (right) are given as visual conditions, respectively.

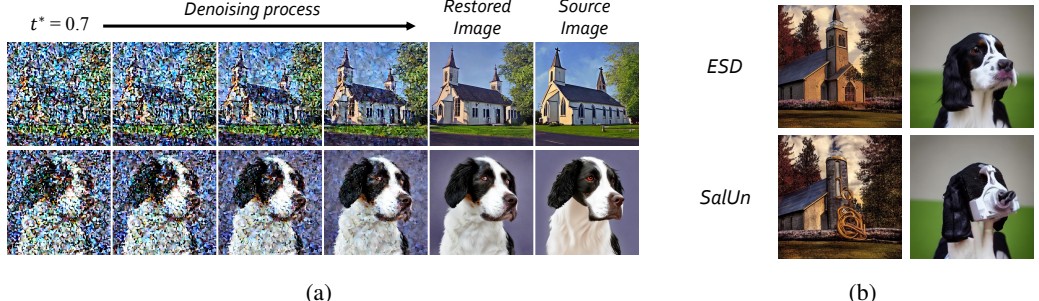

(a)                                                                              (b)

Figure 6: (a) Visualization of concept restoration process. In this example, after injecting noise equivalent to $t^* = 0.7$ to the image, target concept is reconstructed through the denoising process. Original Stable Diffusion with unconditional generation is used. (b) Results of concept restoration using classifier-free guidance, when $t^* = 0.5$. Stable diffusion unlearned with ESD and SalUn are used. More samples can be found in the Appendix H.3.

application scenarios, respectively. For each target concept, we select 50 reference images. We also extract edges from the reference images to convert them into sketches. We generate five images for each condition and evaluate the presence of the target concept in the generated images using GPT-4o. Detailed experimental settings are provided in Appendix G.1.

**Results.** Fig. 5 and Table 6 show the samples generated with visual conditions and the proportion of generated images containing the target concept, respectively. All unlearning methods still generate the target concept with the visual conditions. For example, even after unlearning, the target proportion of the church, remains above 0.9 across all methods, when a sketch is provided. Compared to Table 2, the unlearned model generates the target concepts more with additional visual condition, highlighting a decline in robustness. The result indicates that current unlearning methods are vulnerable in downstream tasks.

> ♀ **Takeaway.** The results of text-to-image unlearning are not consistent in downstream tasks. Especially, additional visual conditions can easily break the unlearned models.

## 6.2 To What Extent Can the Unlearned Model Restore a Target Concept from a Noisy Input?

The results on the downstream tasks raise interesting questions: How robust are unlearning methods in preventing the generation of the target concept under visual conditions? What level of visual context is required to restore the target concept from an unlearned model? To further investigate the robustness of unlearning methods, we utilize the denoising process of the diffusion models.

Choi et al. (2022) have been shown that even when a considerable level of noise is added to the image, diffusion models can still restore a reference image without requiring a holistic context. However, if the model is successfully unlearned, the model should no longer be able to recover the target concept. Through the following experiments, we verify whether the unlearned model restores the target concept from the noisy version of the image with the target concept. Different levels of noise can represent the different levels of model guidance.

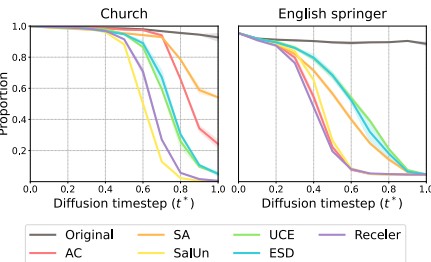

Figure 7: Concept restoration performance for the two concepts, church and English springer.

Table 7: Concept restoration score of unlearning methods.

| | Church | Parachute | Gas pump | English springer | Average |
|---|---|---|---|---|---|
| Original | 0.976 | 0.950 | 0.855 | 0.945 | 0.932 |
| AC | 0.847 | 0.725 | 0.730 | 0.399 | 0.675 |
| SA | 0.889 | 0.787 | 0.627 | 0.528 | 0.708 |
| SalUn | 0.597 | 0.600 | 0.593 | 0.421 | 0.553 |
| UCE | 0.723 | 0.663 | 0.576 | 0.595 | 0.639 |
| ESD | 0.738 | 0.724 | 0.572 | 0.582 | 0.654 |
| Receler | 0.641 | 0.630 | 0.500 | 0.384 | 0.539 |

**Experimental settings.** We adopt the experimental setup from Choi et al. (2022). As shown in Fig. 6a, we first add noise to reference images via a forward diffusion process. We then perform the denoising process to restore the images. The denoising process is conditioned on a simple prompt. We set the step size to 0.02. To conduct a more in-depth analysis, we vary the amount of noise through the forward process time $t^*$ with intervals of 0.1.

For evaluation, we measure the target proportion of the restored images and plot the proportion curve against the different levels of noises $t^*$ for each unlearning method. We then calculate the area under the curve (AUC) to get the concept restoration performance. We use pretrained ImageNet (Deng et al., 2009) classifier with ResNet-50 (He et al., 2016) architecture to classify the restored images. We use the original Stable Diffusion to generate a dataset of 1,000 images per concept. Detailed experimental settings are provided in Appendix H.1.

**Results.** Fig. 6b shows the results of the concept restoration using the unlearned models for concepts church and English springer. In the case of ESD, the features of each target concept are clearly represented, whereas in SalUn, the alternative concept is generated instead. Fig. 7 and Table 7 present the proportion curves and AUCs of the concept restoration, respectively. As shown in Fig. 7, when the target concept is English springer, all methods have a lower target proportion than the church for the same $t^*$ values. This result indicates that the unlearning methods are more robust when the target concept is English springer under given visual conditions. AC, SalUn, and Receler achieve almost zero target proportion at $t^* = 0.6$, indicating strong robustness against visual conditions of this size. In contrast, SA, ESD, and UCE are shown to be less robust under the same visual condition size. Table 7 shows that SalUn and Receler, with lower AUC around 0.5, are more robust to visual conditions than the other methods.

> 💡 **Takeaway.** The concept restoration evaluation provides an in-depth assessment of model robustness on visual conditions with varying semantics. The evaluation can be used as a proxy for downstream tasks.

## 7 CONCLUSION & LIMITATIONS

Our study demonstrates that current unlearning techniques for text-to-image diffusion models are imperfect. While they can partially mitigate the generation of harmful or unwanted content, their application in real-world scenarios remains limited due to issues with robustness, image quality, and unintended side effects. As text-to-image diffusion models continue to evolve, future research needs to focus on addressing these limitations. This includes improving the generalization of unlearning methods to complex prompts, minimizing the trade-off between performance and image quality, and developing techniques to prevent the over-erasing effect. By releasing our comprehensive evaluation framework with the source codes, we hope to inspire further research in this area, leading to more reliable and effective unlearning methods.

**Limitations.** Our benchmark currently focuses on unlearning object-based concepts due to the difficulty of evaluating whether generated images contain specific styles or elements that could raise copyright issues. Legal concerns make this task more complicated. If these challenges are resolved, we believe our benchmark could be expanded to include a wider range of concepts. Also, we do not perform NSFW experiments (Schramowski et al., 2023) to protect the mental health of the researchers, but we believe our findings can be generalized to the NSFW content.

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
