APPENDIX

# A  TEXT-TO-IMAGE DIFFUSION UNLEARNING METHOD

## A.1  ABLATING CONCEPT (AC)

AC (Kumari et al., 2023) uses an alternative concept $c^*$ to prevent the generation of the target concept $c$. The objective is defined as follows:

$$\mathcal{L}_{\text{AC}} = \mathbb{E}_{\boldsymbol{\epsilon}, \mathbf{x}_t, \boldsymbol{c}^*, \boldsymbol{c}, t}[w_t \|\boldsymbol{\epsilon}_\theta(\mathbf{x}_t, \boldsymbol{c}^*, t).\text{sg}() - \boldsymbol{\epsilon}_\theta(\mathbf{x}_t, \boldsymbol{c}, t)\|_2^2], \tag{1}$$

where $w_t$ is a weight of the objective, and .sg() indicates a stop-gradient operator. ACprevents the model from generating the target concept by behaving as if the alternative concept $c^*$ is present when the target concept is given.

## A.2  SELECTIVE AMNESIA (SA)

SA (Heng & Soh, 2024) leverages techniques from continual learning, including Elastic Weight Consolidation (EWC) (Kirkpatrick et al., 2017) and generative replay (GR) (Shin et al., 2017):

$$\mathcal{L}_{\text{SA}} = \mathbb{E}_{q(\mathbf{x}|\mathbf{c})p_f(\mathbf{c})}\left[\|\boldsymbol{\epsilon} - \boldsymbol{\epsilon}_\theta(\mathbf{x}_t, t)\|^2\right] - \lambda \sum_i \frac{F_i}{2}(\theta_i - \theta_i^*)^2 + \mathbb{E}_{p(\mathbf{x}|\mathbf{c})p_r(\mathbf{c})}\left[\|\boldsymbol{\epsilon} - \boldsymbol{\epsilon}_\theta(\mathbf{x}_t, t)\|^2\right], \tag{2}$$

where $q(\mathbf{x}|\boldsymbol{c})$ is a distribution of an alternative concept, and $p(\mathbf{x}|\boldsymbol{c})$ represents a distribution of remaining concepts. For MNIST and Stable Diffusion, SA uses a uniform distribution over the pixel values and a set of samples containing the alternative concept for the $q(\mathbf{x}|\boldsymbol{c})$, respectively.

## A.3  SALIENCY UNLEARNING (SALUN)

SalUn (Fan et al., 2023) utilizes the random labeling (Golatkar et al., 2020), which is widely used for classifier unlearning, for the text-to-image unlearning:

$$\mathcal{L}_{\text{SalUn}} := \mathbb{E}_{(\mathbf{x},\mathbf{c})\sim\mathcal{D}_f, t, \boldsymbol{\epsilon}\sim\mathcal{N}(0,1),\mathbf{c}'\neq\mathbf{c}}\left[\|\boldsymbol{\epsilon}_\theta(\mathbf{x}_t|\mathbf{c}') - \boldsymbol{\epsilon}_\theta(\mathbf{x}_t|\mathbf{c})\|_2^2\right] + \beta\ell_{\text{MSE}}(\theta; \mathcal{D}_r), \tag{3}$$

where $\mathcal{D}_f$ represents the samples of the target concept $\boldsymbol{c}$, and $\mathcal{D}_r$ represents the samples of the remaining concepts. By random forgetting, SalUn prevents the generation of the target concept. Additionally, SalUn uses a saliency map, which is computed by the scale of gradients from the loss $\mathcal{L}_{\text{SalUn}}$, to fine-tune a subset of weights of the diffusion model:

$$\mathbf{m}_{\mathbf{S}} = \mathbf{1}\left(\left|\nabla_\theta \ell_{\text{MSE}}(\theta; \mathcal{D}_f)\Big|_{\theta=\theta_o}\right| \geq \gamma\right), \tag{4}$$

where, $\theta_o$ represents the weights of the pretrained diffusion model and $\gamma$ represents a threshold.

## A.4  ERASED STABLE DIFFUSION (ESD)

ESD (Gandikota et al., 2023) fine-tunes the diffusion model with the following objective function:

$$\mathcal{L}_{\text{ESD}} = \mathbb{E}_{\mathbf{x}_t, t}[\|\boldsymbol{\epsilon}_\theta(\mathbf{x}_t, t) - (\boldsymbol{\epsilon}_{\theta^*}(\mathbf{x}_t, t) - \eta(\boldsymbol{\epsilon}_{\theta^*}(\mathbf{x}_t, \boldsymbol{c}, t) - \boldsymbol{\epsilon}_{\theta^*}(\mathbf{x}_t, t)))\|_2^2], \tag{5}$$

where $\theta$ represents the trainable parameters of the diffusion model, $\theta^*$ represents the fixed original diffusion model, and $c$ represents the target concept. The modified score function shifts the data distribution to reduce the probability of generating images containing the target concept $\boldsymbol{c}$.

## A.5  UNIFIED CONCEPT EDITING (UCE)

UCE (Gandikota et al., 2024) edit weights of cross-attention layers for its unlearning:

$$\min_W \sum_{c_i \in E} \|Wc_i - v_i^*\|_2^2 + \sum_{c_j \in P} \|Wc_j - W^{\text{old}}c_j\|_2^2, \tag{6}$$

where $W$, $W^{\text{old}}$, $E$, and $P$ represent new weights, old weights, concepts to be erased, and concepts to be preserved, respectively. UCE finds the target value $v_{i*} = W^{\text{old}}c_{i*}$ of destination embedding $c_{i*}$ that can prevent the generation of the target concept. A solution of the objective can be calculated in close-form:

$$W = \left( \sum_{c_i \in E} v_i^* c_i^T + \sum_{c_j \in P} W^{\text{old}} c_j c_j^T \right) \left( \sum_{c_i \in E} c_i c_i^T + \sum_{c_j \in P} c_j c_j^T \right)^{-1}. \tag{7}$$

The destination embedding for the object unlearning is equal to a null embedding (*i.e.*, "").

### A.6 RECELER

`Receler` (Huang et al., 2023) trains an adapter-based eraser $E$ using the same objective with ESD. Additionally, `Receler` utilizes masking-based regularization loss to ensure that the eraser can remove only the target concept to be erased.

## B EXPERIMENTAL SETTING DETAILS

We use Stable Diffusion v1.4 as our pretrained models for all experiments. To unlearn the Stable Diffusion, We follow the provided code and instructions on each Github page to train their models accordingly. All of the hyperparameters of the fine-tuning are the same as the values used in the original implementations, except for the implementation of `AC` and the number of epochs of `SA` (Heng & Soh, 2024). Since the original 200 epochs of `SA` are insufficient to achieve effective unlearning for church, we increase the number of epochs to 300.

### B.1 IMPLEMENTATION OF AC

In the original implementation of `AC`, the authors evaluate `AC` only on adjective-like concepts, such as "*Grumpy*" from "*Grumpy cat*". We extend the settings of the `AC` to enable object unlearning. Specifically, given a target concept $c$, we select an alternative concept $c^*$ from a different object and train the diffusion models using the objective function described in Appendix A.1. Although this approach has not been explored in the original research, `AC` can be effectively applied to object unlearning and is categorized as a mapping-based method, similar to `SA` and `SalUn`.

### B.2 EXPERIMENTAL SETTINGS ON MNIST

To evaluate the bias arising from unlearning, we train a conditional diffusion model on the MNIST dataset. following the work of `SA` (Heng & Soh, 2024). The conditional diffusion model uses a UNet architecture and is trained with an objective used in classifier-free guidance (Ho & Salimans, 2022), also applied in Stable Diffusion. This setup enables both unconditional and conditional generation with the same network.

**Unlearning settings.** We fine-tune the DDPM trained on the MNIST dataset using the unlearning methods, including `SA`, `SalUn`, `ESD`, and `AC`. For `SA` and `SalUn`, we adopt the objective functions and hyperparameters from their original implementations on CIFAR-10, which still perform well on the MNIST dataset. For `ESD` and `AC`, we fine-tune the trained DDPM using the same objective as in Stable Diffusion. Specifically, `ESD` is trained for 20,000 iterations with a batch size of 128, similar to `SA`. `AC` is trained for 1,000 iterations with the same batch size, as we observe rapid convergence. Note that we confirm that all methods effectively unlearn the diffusion model.

**Alternative class.** `SA`, `SalUn`, and `AC` requires an alternative class for unlearning. We follow the setting from their original implementation for `SA` and `SalUn`. `SA` uses a uniform distribution over the pixel values. `SalUn` uses a random forgetting algorithm. We observe that `SalUn` generates a single alternative class after unlearning. For `AC`, we select one as the alternative class, as it can be distinguished clearly. When one is the target class, the alternative class is set to zero. Samples of conditional generation of the target class are provided in Appendix F.2.

# C EFFECTIVENESS ON TARGET CONCEPT

## C.1 CLASSIFICATION USING GPT-4O

In most of our experiments, we use GPT-4o to classify whether an image contains the specified concept. The instructions used for classification are provided in Fig. 8.

---

You will be shown an AI-generated image. Your task is to determine whether the bird is clearly visible and accurately depicted.

1. If the {concept} is clearly visible and depicted correctly, respond with "Yes."

2. If the {concept} is not visible, unclear, or inaccurately depicted, respond with "No."

---

Figure 8: Instructions used to classify images with GPT-4o, where "concept" is replaced by the specific concept being classified.

## C.2 DETAIL OF PROMPT CURATION

For the prompt curation process, we use GPT-4o to generate prompts. We explicitly ask GPT-4o to generate prompts in multiple languages, including French, Spanish, and Italian. The instruction used to generate diversified prompts is shown in Fig. 9. Finally, we manually inspect the quality of the collected candidates and curate the final 100 prompts for each concept. Randomly sampled prompts from the final curated set are shown in Table 9. Roughly 15% of the final prompts are written in a language other than English.

---

Write 20 Stable Diffusion prompts to generate images depicting {concept}. Consider the following requirements:

1. You may use synonyms for {concept}

2. The prompt should be clear, high-quality, and depict the exterior of the {concept} (you can use various trigger words like 8k, HD, hyper-detailed, 35mm film grain, etc.).

3. You may write the prompt in different languages.

4. The prompts should vary in terms of length and expression.

---

Figure 9: Instruction used to generate prompts for the target concept.

**Compare the diverse prompt with other methods.** We compare the variety of prompts used in our approach with those generated by other methods. As shown in Table 8, our approach generates a larger number of prompts while also incorporating multiple languages. Furthermore, our prompts are longer on average, as indicated by the mean prompt length, suggesting that they are more complex and sophisticated than those generated by other methods.

| | # of prompt | LLM | Other language | Mean prompt length |
|---|---|---|---|---|
| AC | 10 | ✔ | | 7.35 |
| Receler | 50 | ✔ | | 14.81 |
| Ours | 100 | ✔ | ✔ | 20.25 |

Table 8: Comparison of prompt for each method

| concept | prompt |
| --- | --- |
| church | "*Church Surrounded by Cherry Blossoms in Spring: A traditional church with a cross on its roof, surrounded by cherry blossom trees in full bloom. Rendered in 4k with vibrant pinks and whites, capturing the delicate flowers in the foreground.*"
"*Under a clear, starry sky, a small rural church glows softly from within, surrounded by fields of tall grass that sway gently in the night breeze. Ultra HD 8k, with intricate details of the church.*"
"*Majestuosa catedral que se alza sobre una metrópolis en expansión", detalles góticos intrincados, el corazón y alma de la ciudad, Canon EOS-1D X Mark III, hora dorada con sombras contrastantes, estilo arquitectónico en ultra alta definición.*" |
| parachute | "*Parachutist drifting above an alpine landscape, snow-capped peaks in the background, ultra-realistic, richly textured*"
"*Skydiver descending with an open parachute, sharp details and vivid sky, no clouds in sight.*"
"*Un parachute s'ouvrant à haute altitude, tissu gonflé, cordes détaillées, perspective dynamique, hyper-réaliste, ultra HD.*" |
| gas pump | "*8k hyper-detailed image of a gas station during sunset, orange hues casting long shadows, gas pumps reflecting the warm light, a calm atmosphere.*"
"*A modern gas pump with an integrated charging port for electric vehicles, hyper-detailed, HD, clean and futuristic design.*"
"*Imagen ultra-detallada en 8K de una gasolinera con múltiples bombas de gasolina en fila, sus columnas metálicas brillando bajo un sol brillante del mediodía.*" |
| English springer | "*With a stick in its mouth, the English Springer bounds through tall grass, ultra HD, dynamic fur movement, richly textured meadow.*"
"*English Springer Spaniel sitting calmly in a field of daisies, highly detailed fur, bright sunlight, clear blue sky in the background.*"
"*Primer plano de la oreja y el pelaje de un Springer Spaniel Inglés, HD, textura muy detallada, enfoque intenso, iluminación natural.*" |

Table 9: Examples of diverse prompts for each concept.

## C.3 QUALITATIVE RESULT

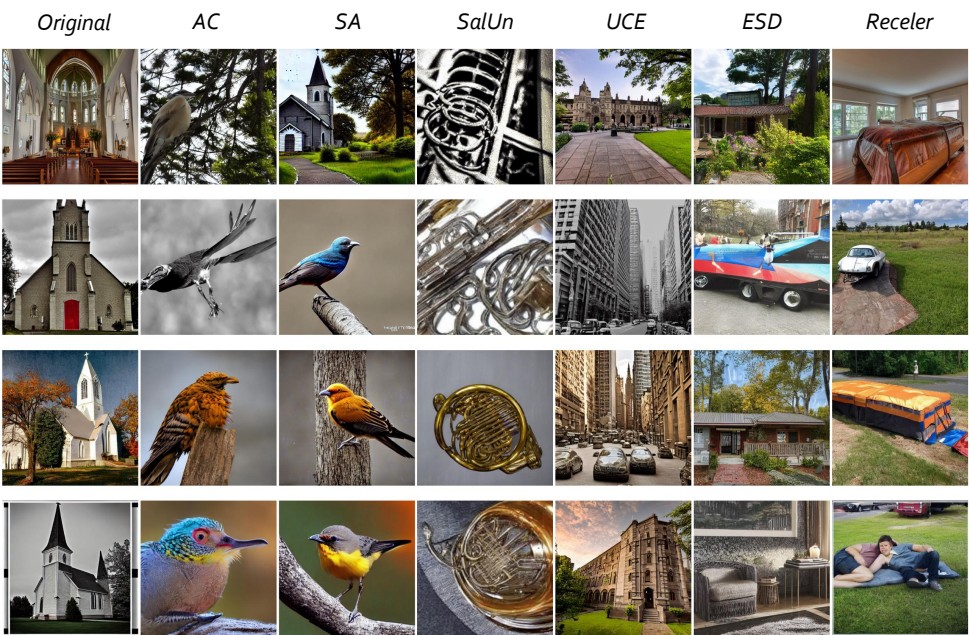

Figure 10: church images generated from simple prompt.

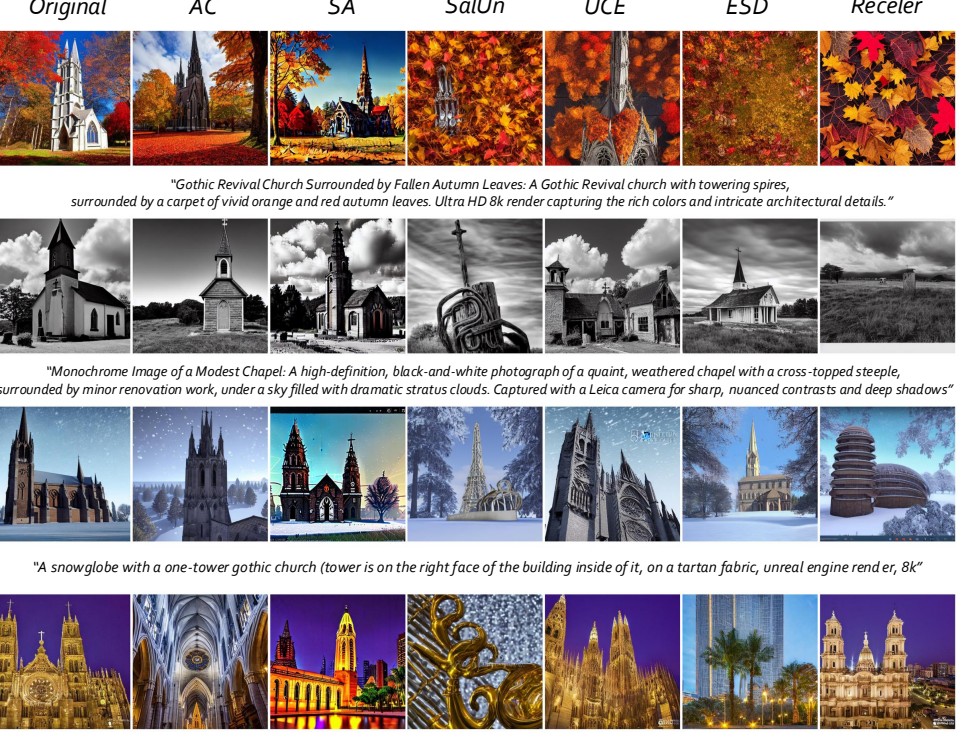

Figure 11: church images generated from diverse prompt.

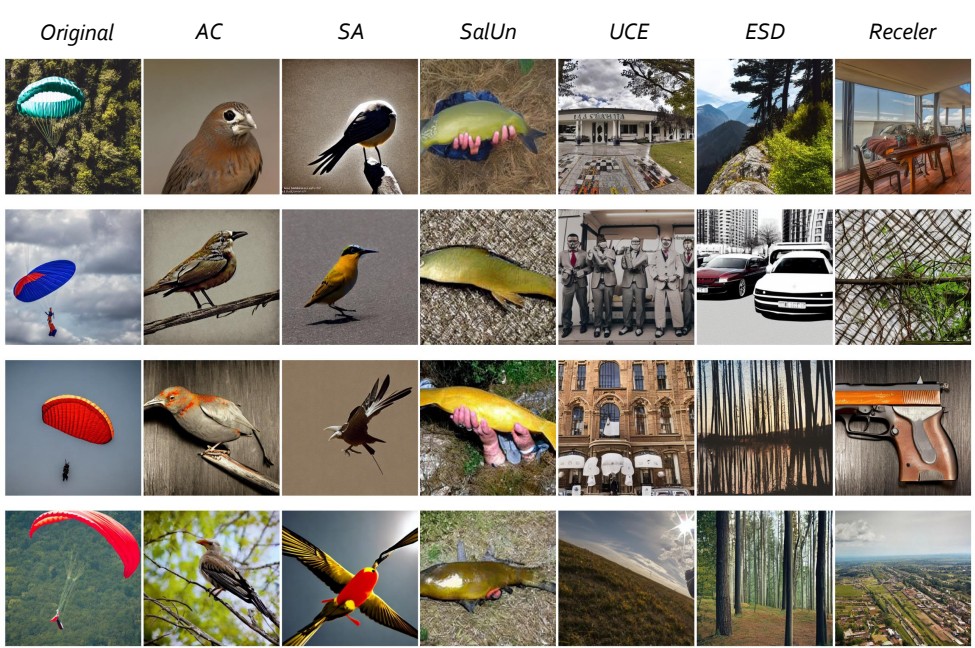

Figure 12: parachute images generated from simple prompt.

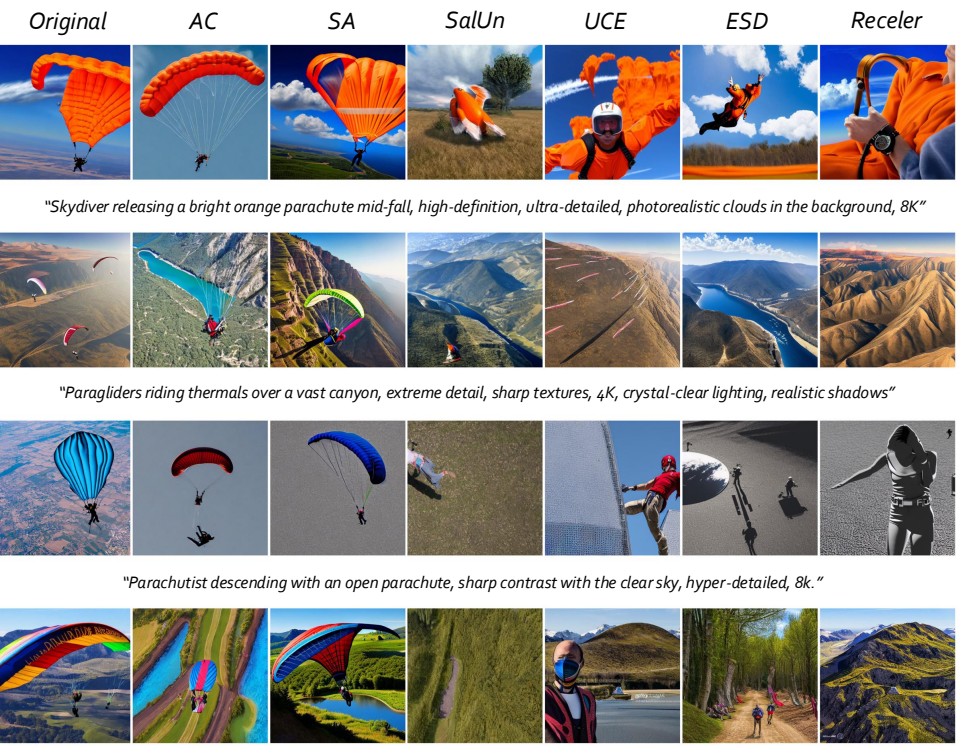

*"Skydiver releasing a bright orange parachute mid-fall, high-definition, ultra-detailed, photorealistic clouds in the background, 8K"*

*"Paragliders riding thermals over a vast canyon, extreme detail, sharp textures, 4K, crystal-clear lighting, realistic shadows"*

*"Parachutist descending with an open parachute, sharp contrast with the clear sky, hyper-detailed, 8k."*

*"Une personne en parapente avec un parachute à motifs sur un paysage vallonné, ultra-clair, richement détaillé, couleurs vives, 8k."*

Figure 13: parachute images generated from diverse prompt.

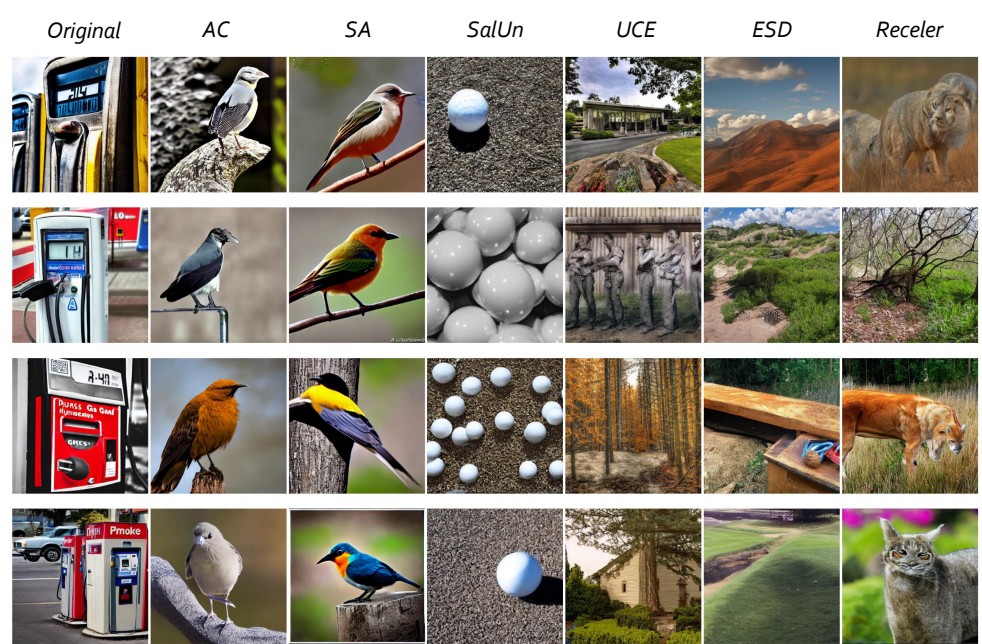

Figure 14: gas pump images generated from simple prompt.

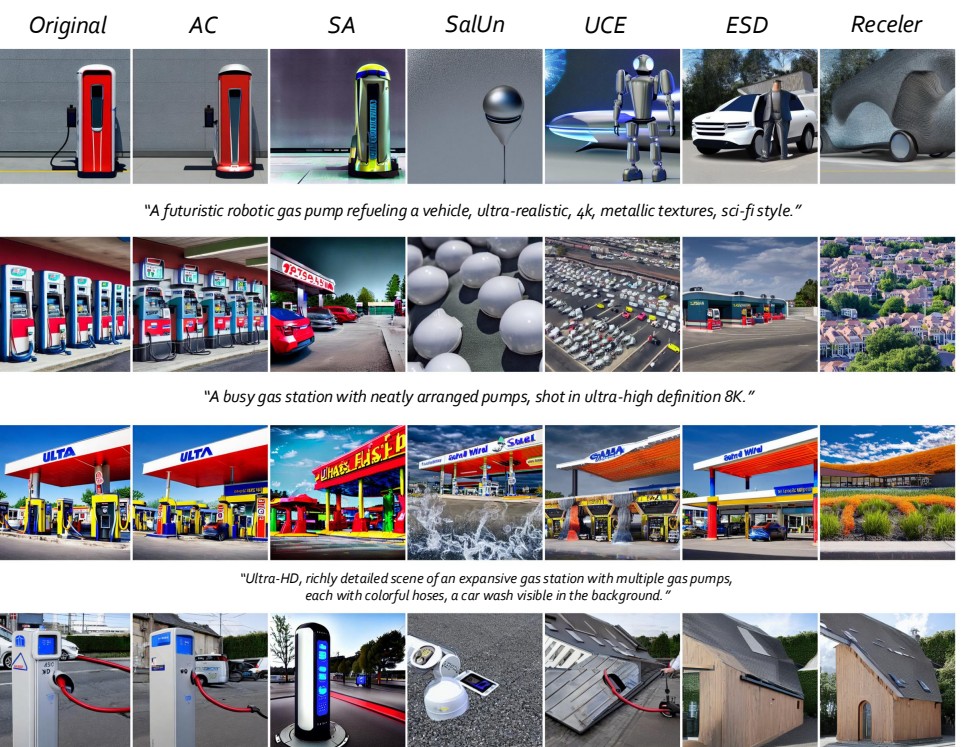

Figure 15: gas pump images generated from diverse prompt.

Original    AC    SA    SalUn    UCE    ESD    Receler

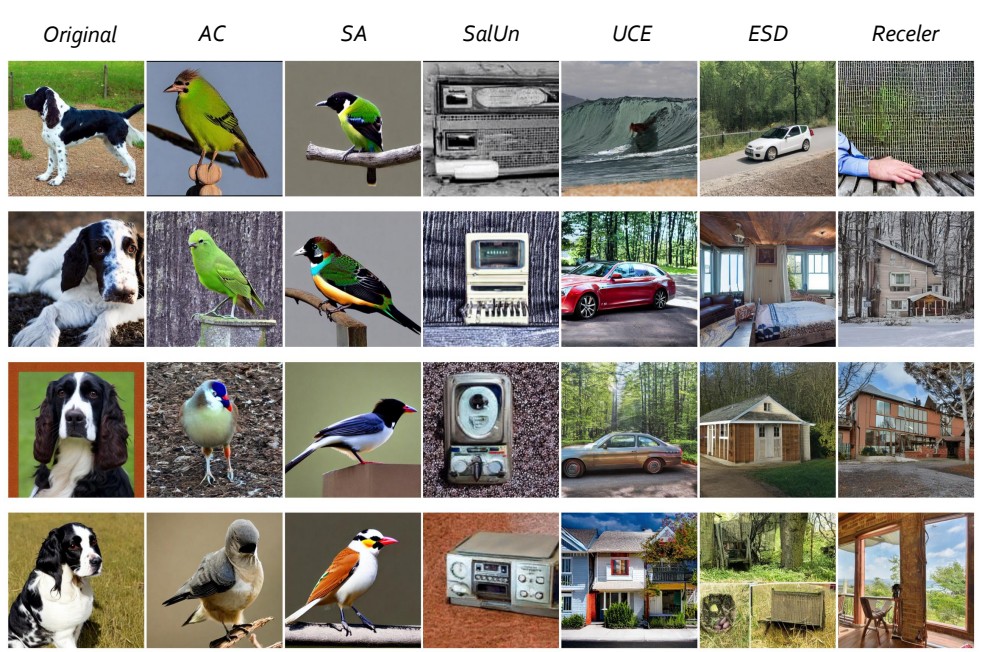

Figure 16: English springer images generated from simple prompt.

Original    AC    SA    SalUn    UCE    ESD    Receler

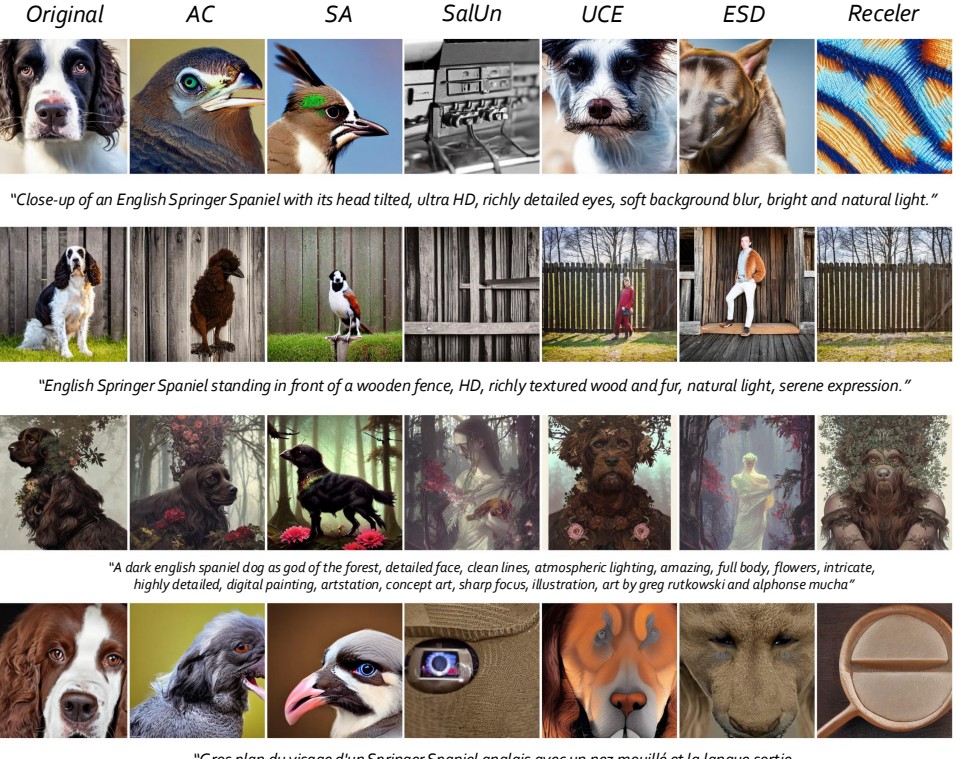

"Close-up of an English Springer Spaniel with its head tilted, ultra HD, richly detailed eyes, soft background blur, bright and natural light."

"English Springer Spaniel standing in front of a wooden fence, HD, richly textured wood and fur, natural light, serene expression."

"A dark english spaniel dog as god of the forest, detailed face, clean lines, atmospheric lighting, amazing, full body, flowers, intricate, highly detailed, digital painting, artstation, concept art, sharp focus, illustration, art by greg rutkowski and alphonse mucha"

"Gros plan du visage d'un Springer Spaniel anglais avec un nez mouillé et la langue sortie, hyper-détaillé, résolution 8k, texture naturelle de la peau, expression joyeuse et brillante."

Figure 17: English springer images generated from diverse prompt.

## D   SELECTIVE ALIGNMENT TASK

We propose a selective alignment task to assess whether the unlearned model can accurately generate the remaining concepts within a given prompt. To do this, we curate ten distinct background concepts and generate corresponding images. We provide the prompt that is used in the selective alignment tasks in Table 10.

| Prompt |
|---|
| "A {concept} *with cherry blossoms.*" |
| "A {concept} *with a star-field night sky.*" |
| "A {concept} *with a beach.*" |
| "A {concept} *with snow.*" |
| "A {concept} *with a sunset.*" |
| "A {concept} *with fallen leaves.*" |
| "A {concept} *with a desert.*" |
| "A {concept} *with a meadow.*" |
| "A {concept} *with a flower field.*" |
| "A {concept} *with a forest.*" |

Table 10: Prompts used in the selective alignment task, where {concept} is replaced by the target concept.

### D.1   QUALITATIVE RESULTS

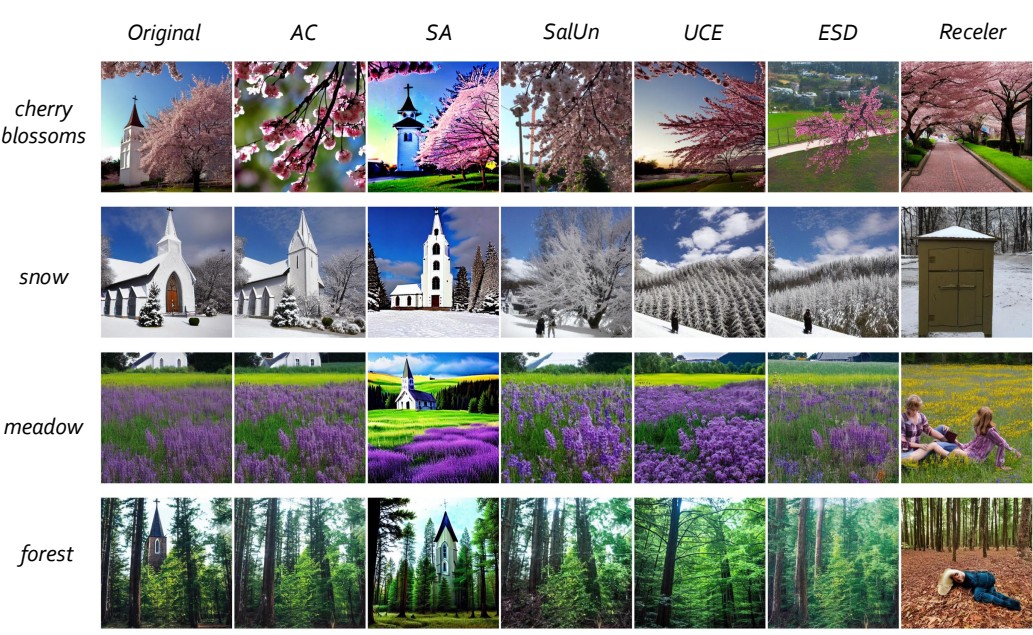

Figure 18: Samples of generated images with background from church-erased model for selective alignment task

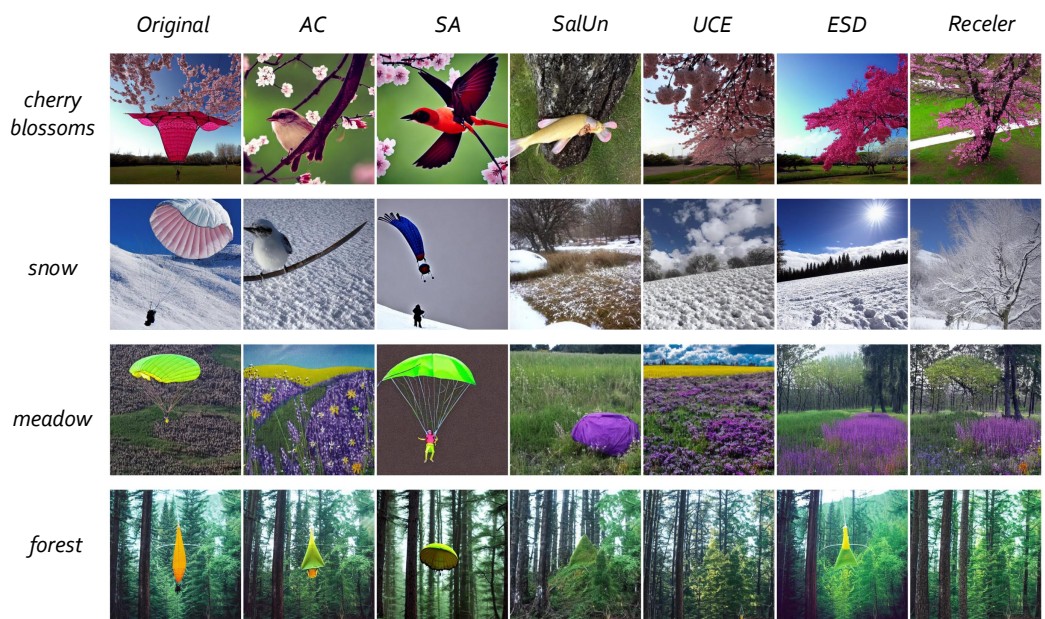

Figure 19: Samples of generated images with background from parachute-erased model for selective alignment task

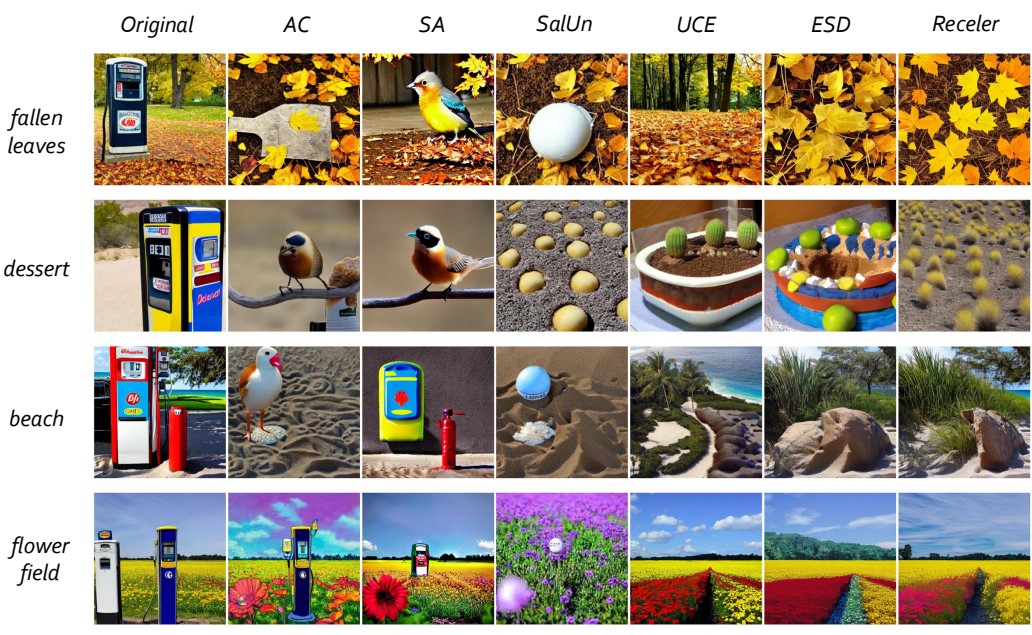

Figure 20: Samples of generated images with background from gas pump-erased model for selective alignment task

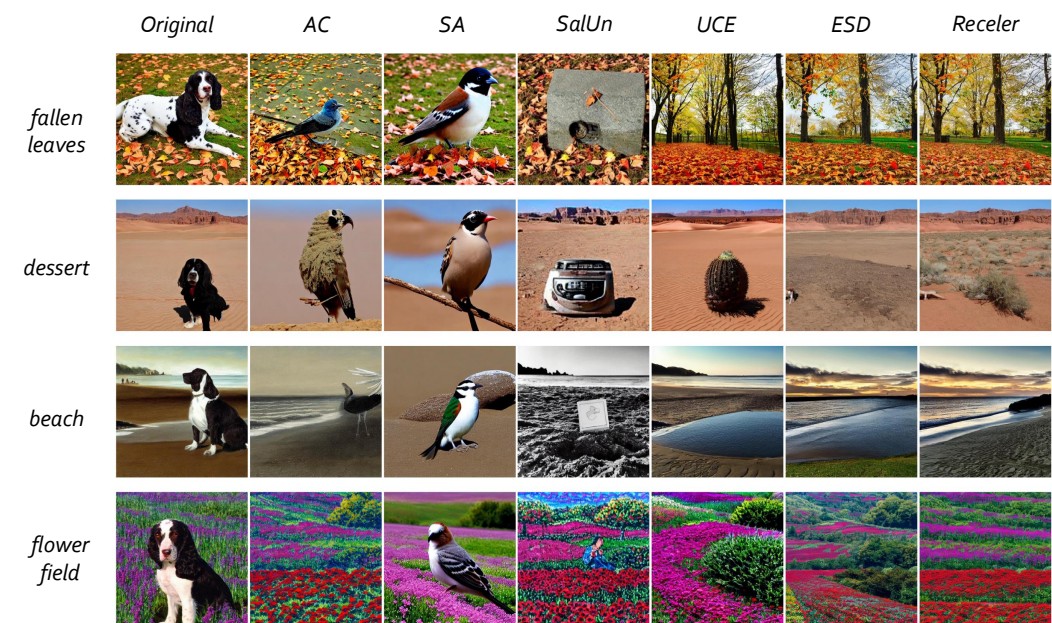

Figure 21: Samples of generated images with background from English springer-erased model for selective alignment task

# E  OVER-ERASING EFFECT

## E.1  RELATED CONCEPTS

In Section 5.1, we discuss the issue of over-erasing, where the unlearned model removes the target concept and fails to generate its related concepts. To evaluate this effect, we select five related concepts for each target concept. The categories and their corresponding related concepts are provided in Table 11.

| Concept | Category | Related concept |
|---|---|---|
| church | Christian | cross, alter, pulpit, rosary, bible |
| parachute | Flying object | air balloon, jet, kite, drone, aircraft |
| gas pump | Machine | vending machine, ATM, slot machine, gumball machine, coffee machine |
| English springer | Dog breed | Saint Bernard, Beagle, Chihuahua, Shiba Inu, Samoyed |

Table 11: Related concept and its category for each target concept.

E.2 QUALITATIVE RESULTS

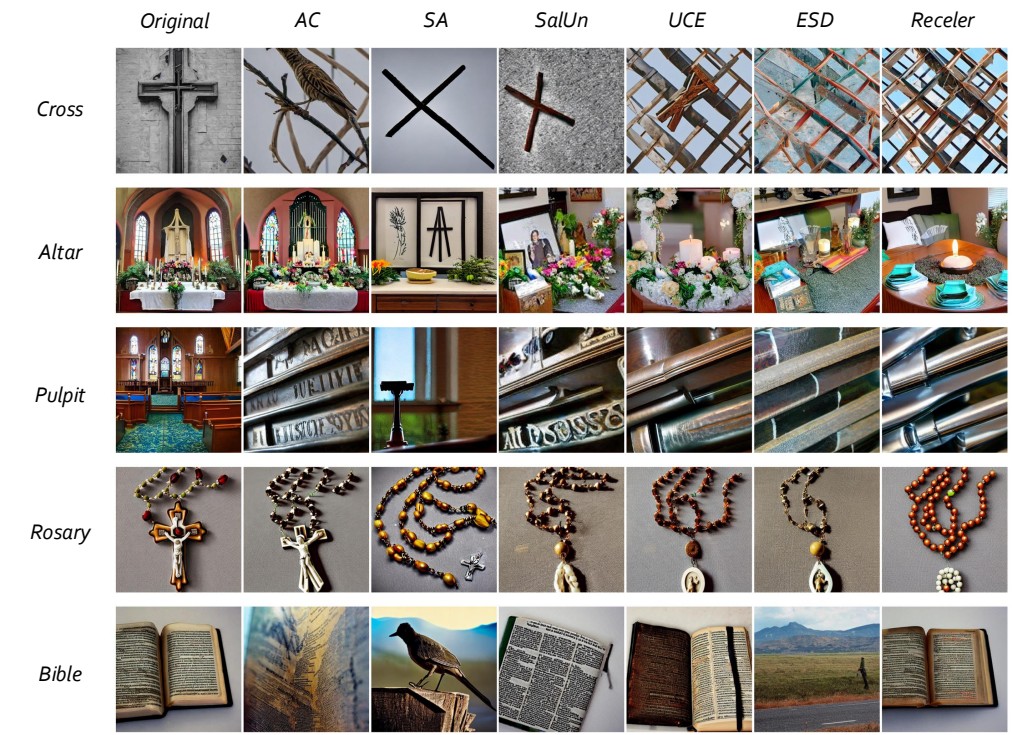

Figure 22: Samples of generated images that related to church.

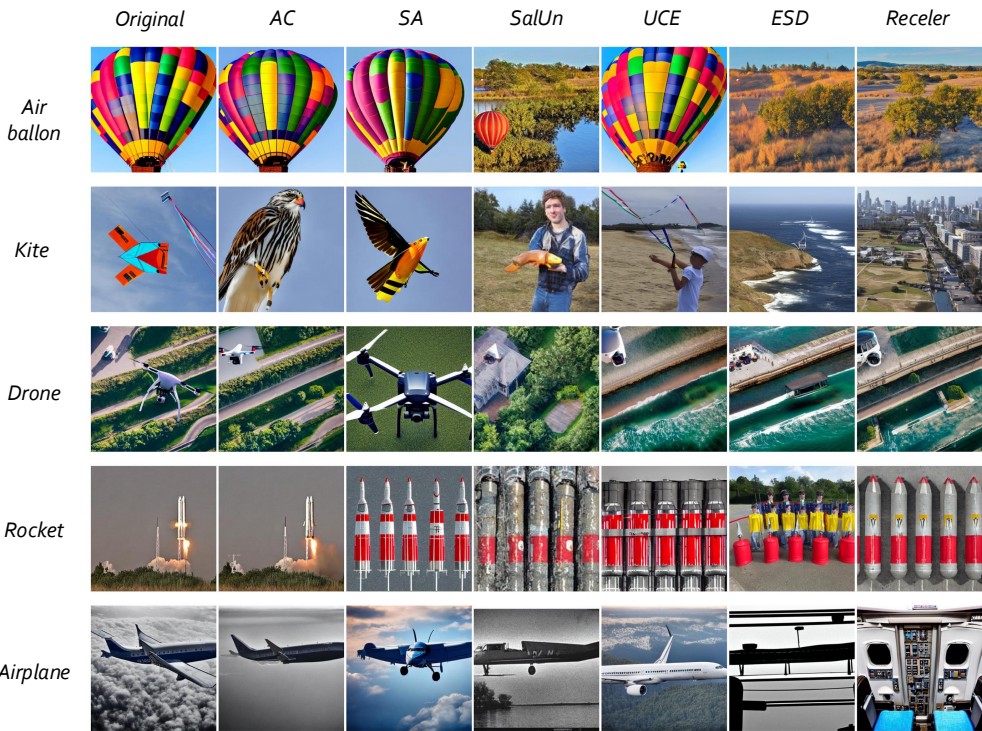

Figure 23: Samples of generated images that related to parachute.

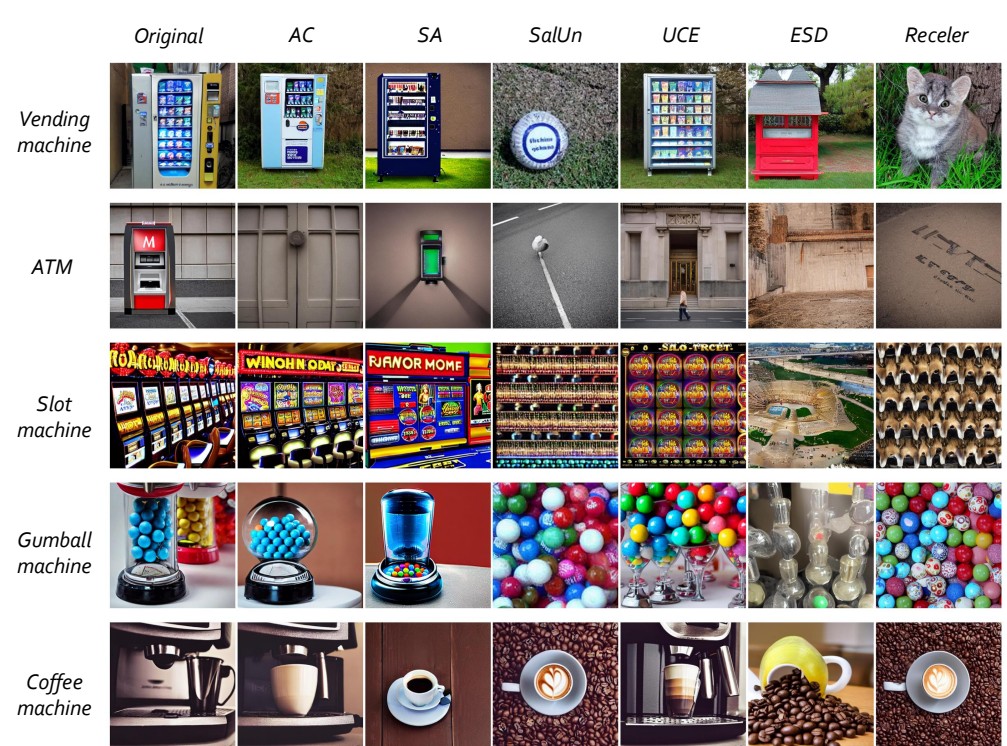

Figure 24: Samples of generated images that related to gas pump.

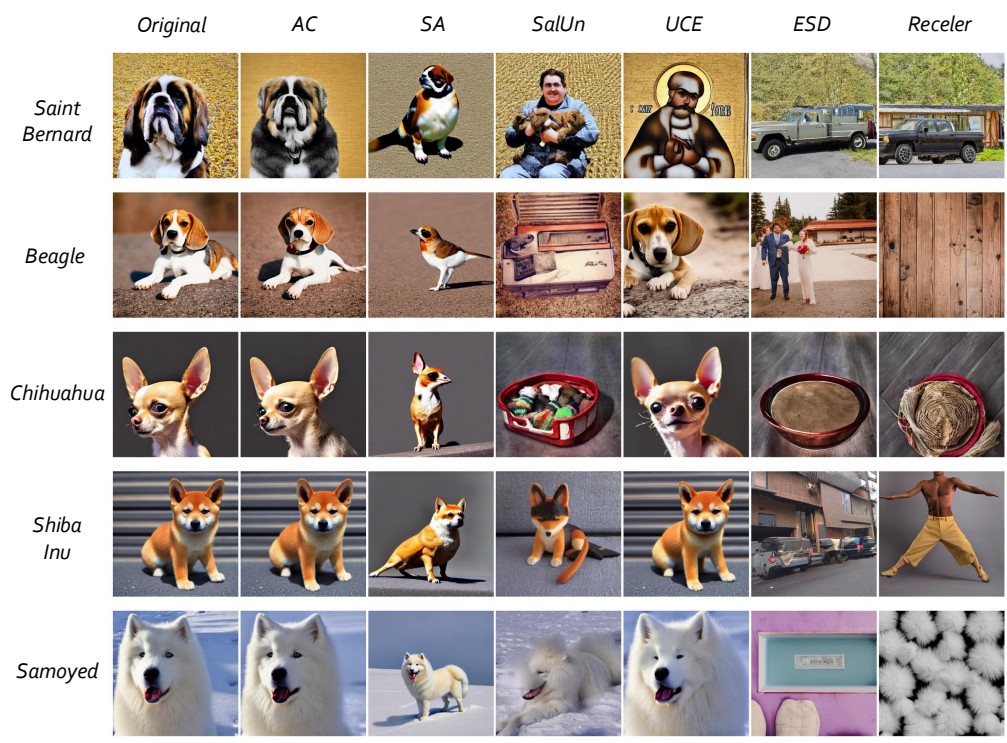

Figure 25: Samples of generated images that related to English springer.

# F BIAS: UNCONDITIONAL GENERATION

## F.1 BIAS IN STABLE DIFFUSION

In Section 5.2, we observe that unlearning could cause bias. This raises a question of whether bias also occurs in Stable Diffusion. We find that the mapping-based methods, AC and SA, also introduce bias when these methods unlearn the Stable Diffusion. To check the bias on Stable Diffusion, we generate 500 images from Stable Diffusion unlearned with AC and SA, and we calculate the proportion of generated images containing target concept (*i.e.*, Bird) using GPT-4o. We also calculate the target concept proportion of the original Stable Diffusion.

Table 12 shows the proportion of the target concept unconditionally generated from Stable Diffusion. Roughly 64.5% and 7.6% of images generated from SA and AC, respectively, belong to the alternative class. The feature space of Stable Diffusion is vast, indicating that the models unlearned using SA and AC are significantly biased. The images of the alternative concepts generated through unconditional generation can be found in the Fig. 28.

|  | Original | AC | SA |
| --- | --- | --- | --- |
| Replacement | 0.002 | 0.076 | 0.645 |

Table 12: Proportion of images containing the target concept unconditionally generated from Stable Diffusion. The target concept of AC and SA is bird.

## F.2 QUALITATIVE RESULTS OF CONDITIONAL GENERATION ON MNIST

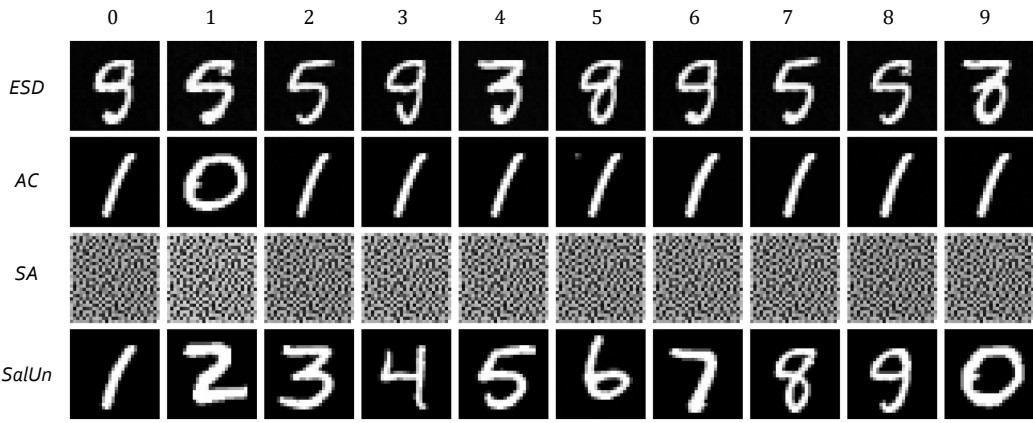

Figure 26: Samples generated with the target concept on the MNIST dataset. Each column shows the target class.

### F.3 QUALITATIVE RESULTS OF UNCONDITIONAL GENERATION ON MNIST

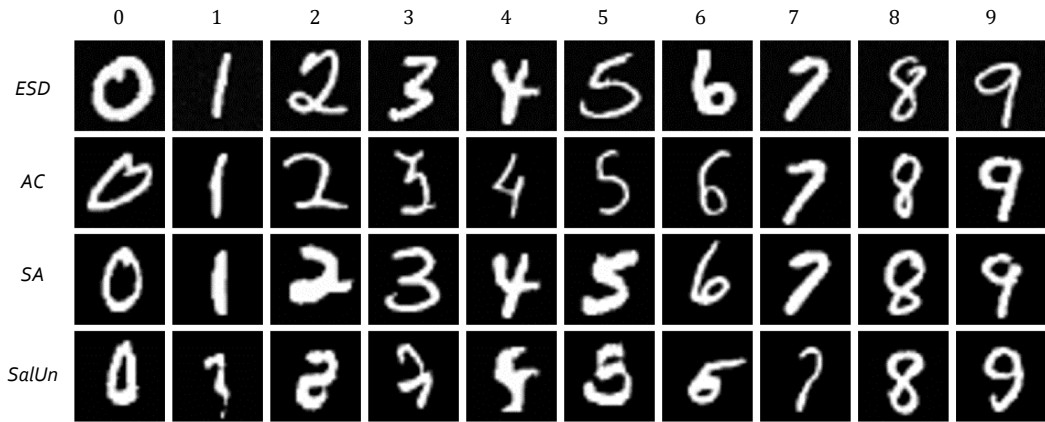

Figure 27: Samples of unconditional generation with an unlearned diffusion model on the MNIST dataset. Each column shows the target class for unlearning, with each image representing a sample classified into that class.

### F.4 QUALITATIVE RESULTS OF UNCONDITIONAL GENERATION IN STABLE DIFFUSION

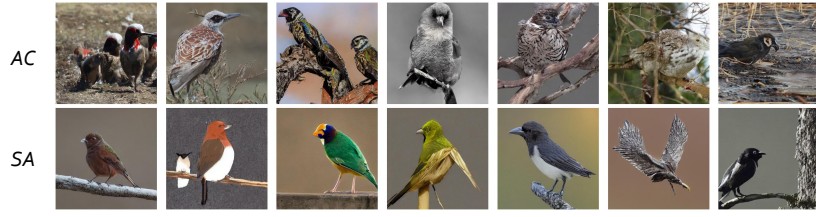

Figure 28: Samples unconditionally generated from Stable Diffusion unlearned with AC and SA. This samples contains the object bird, which is a target concept of both AC and SA.

## G APPLICATION

### G.1 EXPERIMENTAL SETTINGS.

We employ ControlNet (Zhang et al., 2023a) with HED (Xie & Tu, 2015) and ControlNet reference-only (Zhang et al., 2023a). We use the ControlNet v1.1[2] for the sketch-to-image generation. Although the ControlNet is trained with Stable Diffusion v1.5, we find that the ControlNet also works well with Stable Diffusion v1.4. For the ControlNet reference-only, we use the implementation of Diffusers (von Platen et al., 2022)[3].

For each concept, we select 50 reference images. We also extract edges from the reference images to convert them into sketches. We generate five images for each condition and evaluate the presence of the target concept in the generated images using GPT-4o. For this experiment, we additionally use a negative prompt with words that specify the quality of generated images. Specifically, we use "*a photo of {concept}, best quality, HD, extremely detailed, realistic*" for the positive prompt and "*monochrome, lowres, bad quality, bad anatomy, worst quality, low quality, low res, blurry, distortion*" for the negative prompt. We use a UniPC scheduler (Zhao et al., 2024) with a step size of 20.

---

[2] https://huggingface.co/lllyasviel/control_v11p_sd15_softedge
[3] https://github.com/huggingface/diffusers/blob/main/examples/community/stable_diffusion_controlnet_reference.py

## G.2 QUALITATIVE RESULTS.

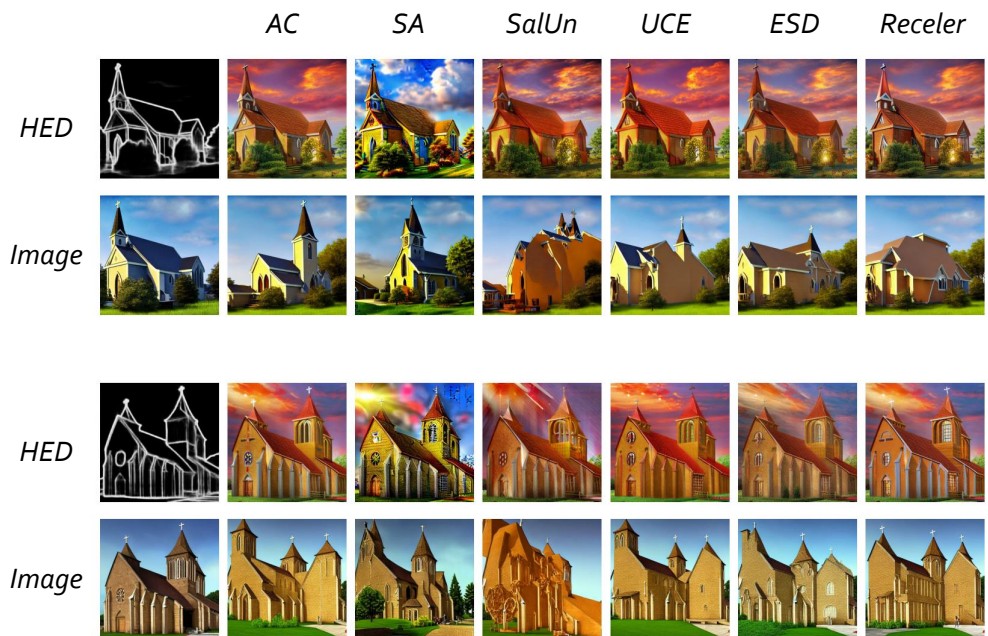

Figure 29: church images generated with additional conditions. The first row consists of images generated with HED, and the second row consists of images generated with an reference image.

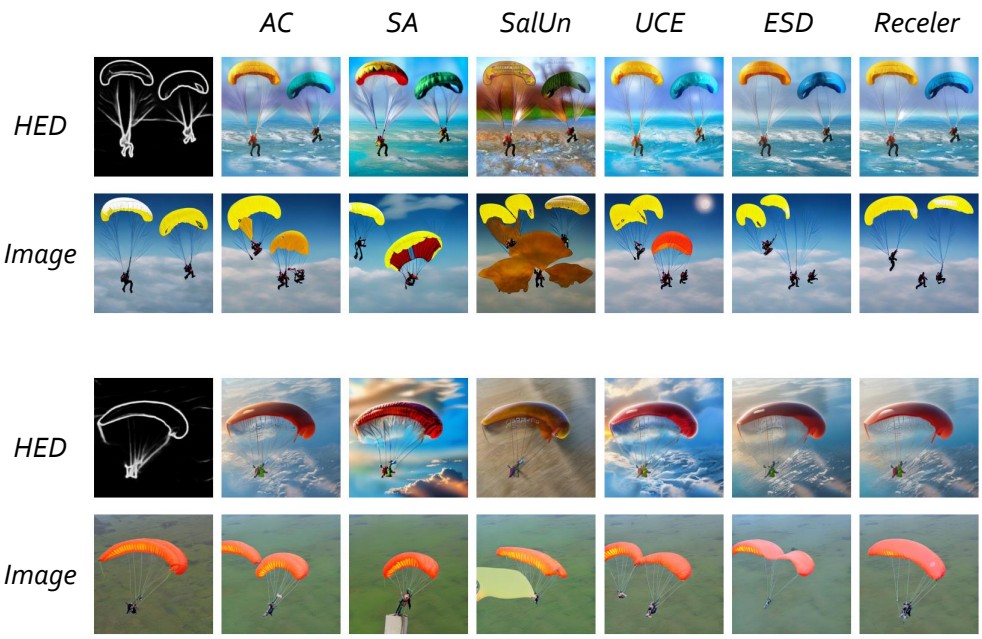

Figure 30: parachute images generated with additional conditions. The first row consists of images generated with HED, and the second row consists of images generated with an reference image.

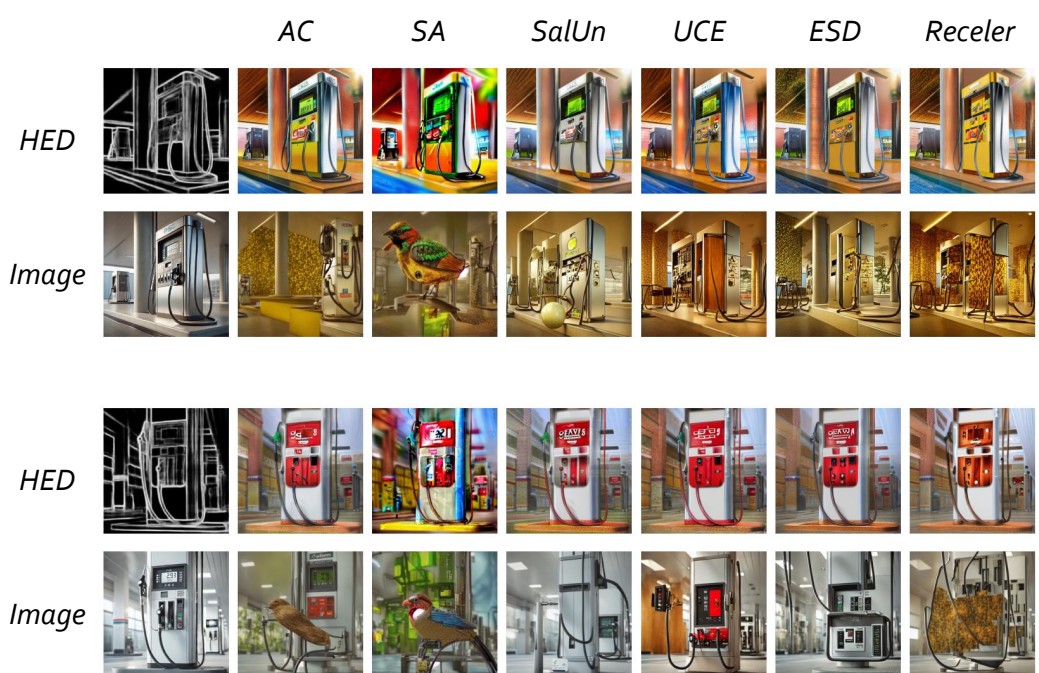

Figure 31: gas pump images generated with additional conditions. The first row consists of images generated with HED, and the second row consists of images generated with an reference image.

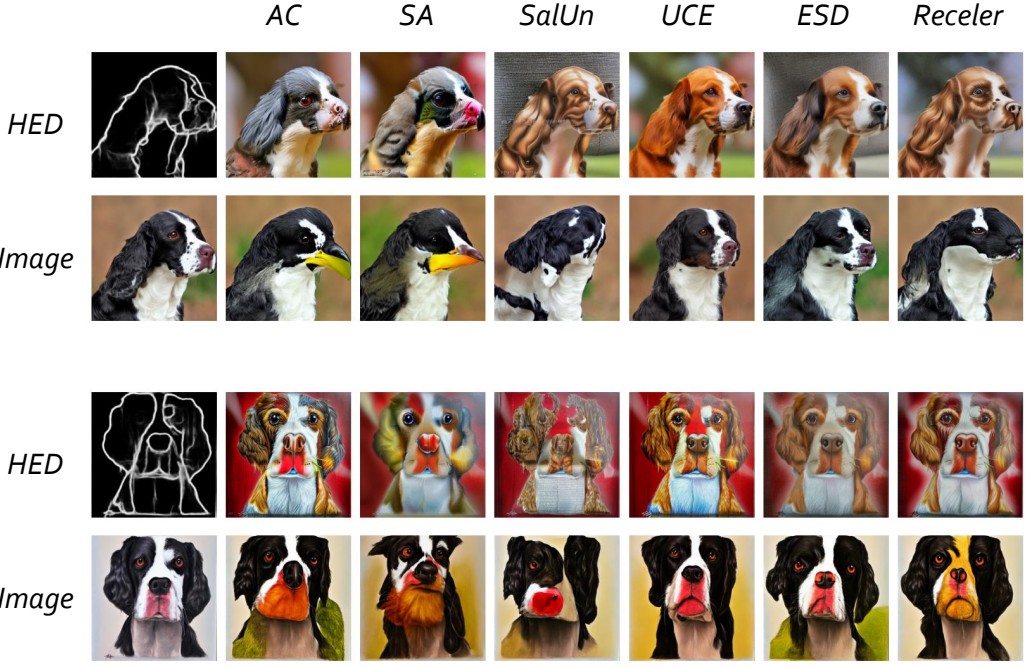

Figure 32: English springer images generated with additional conditions. The first row consists of images generated with HED, and the second row consists of images generated with an reference image.

# H CONCEPT RESTORATION

## H.1 EXPERIMENTAL SETTINGS

To restore the images from noisy reference images, we use original or unlearned Stable Diffusion. First, we add noise to the reference images with a predefined diffusion timestep $t^*$. And then, we perform the denoising process with PNDM scheduler (Liu et al., 2022) and a step size of 0.02. The number of steps in the denoising process can vary depending on the diffusion timestep $t^*$ with intervals of 0.1. We use a simple prompt "*a photo of a {concept}*" for the text condition during the denoising process.

For evaluation, we first classify the recovered images with a pretrained classifier, and we compute the proportion of images containing the target concept. Then, we plot the proportion measured for different $t^*$ values, the same unlearning method, and the same target concept as a curve, and we calculate the area under the curve (AUC) of this curve. We use 1,000 images per concept for the reference images, and we use a pretrained ImageNet classifier with resnet-50 architecture. We conduct the same experiment for all unlearning methods and target concepts.

## H.2 AUCS OF CONCEPT RESTORATION

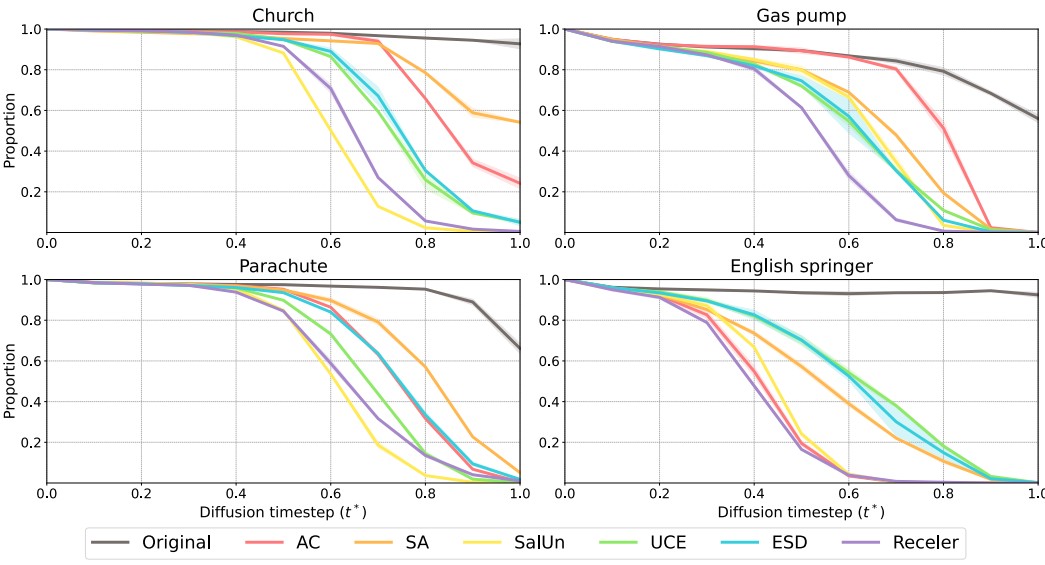

Figure 33: Area under the curves of the concept restoration.

## H.3 QUALITATIVE RESULTS.

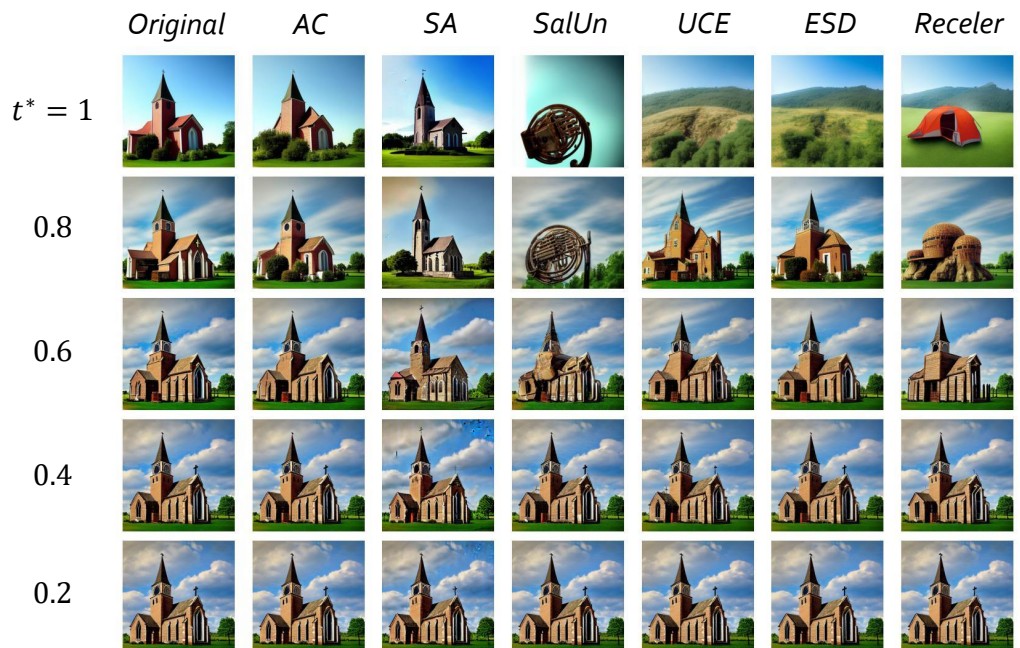

Figure 34: Concept restoration results when the target concept is church. Each row represents the start diffusion timestep $t^*$.

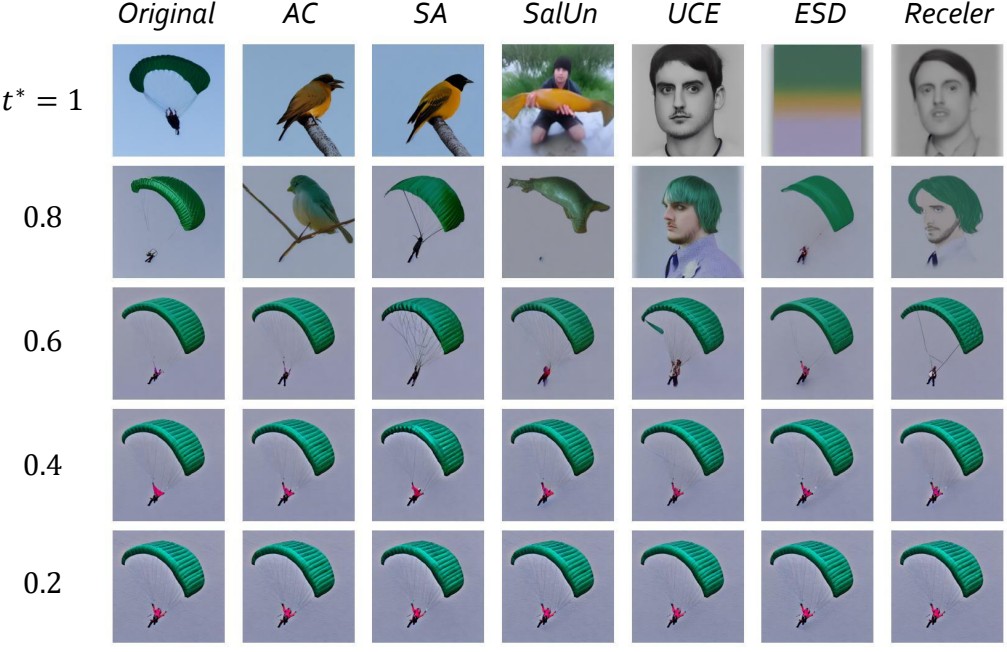

Figure 35: Concept restoration results when the target concept is parachute. Each row represents the start diffusion timestep $t^*$.

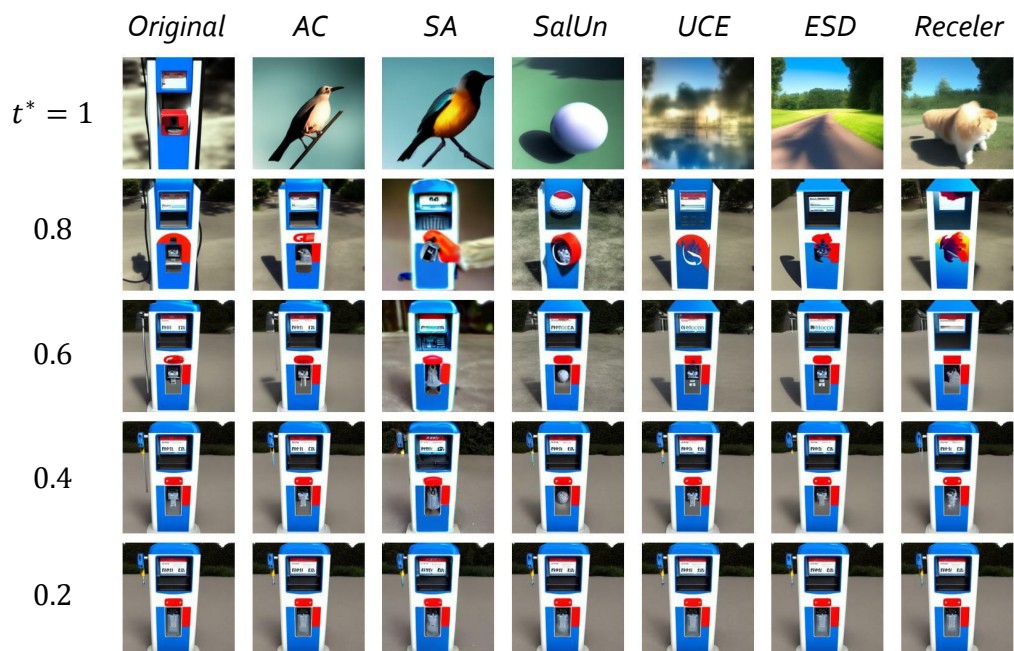

Figure 36: Concept restoration results when the target concept is gas pump. Each row represents the start diffusion timestep $t^*$.

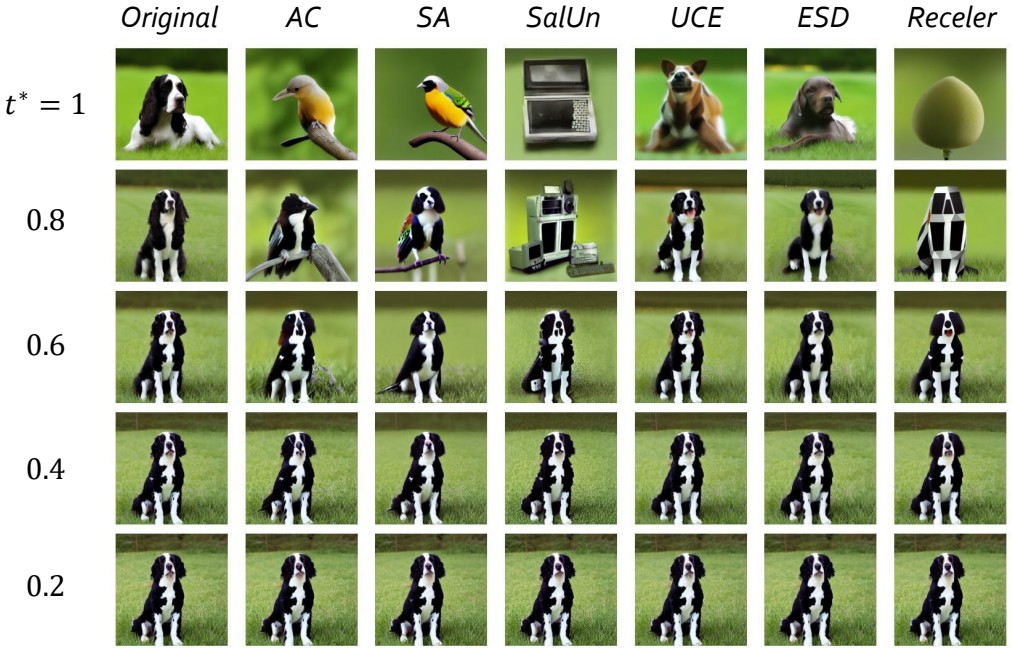

Figure 37: Concept restoration results when the target concept is English springer. Each row represents the start diffusion timestep $t^*$.