# OpenReview forum: "Holistic Unlearning Benchmark: A Multi-Faceted Evaluation for Text-to-Image Diffusion Model Unlearning"
_ICLR.cc/2025/Conference — ICLR 2025 Conference Withdrawn Submission_

### Official Review · Reviewer_bcXg · 2024-11-01

**Soundness:** 4
**Presentation:** 4
**Contribution:** 2
**Rating:** 5
**Confidence:** 3

**Summary:**

Various unlearning techniques have emerged to prevent generating images related to copyright or NSFW content. Although previous methods also has been evaluated, their evaluations were limited to specific scenarios and lacked comprehensive analysis of various side effects. This paper conducts extensive research to uncover the limitations of these methods.

**Strengths:**

- The observation that existing unlearning methods perform differently depending on the complexity of prompts used to test concept removal is insightful.
- The paper suggests several ways to measure model performance more precisely, which contribute valuable analyses.
  - The measurement of influence on related concepts in Section 5.1 appears to be a useful approach for thoroughly testing the over-erasing issue in unlearning.
  - Section 5.2’s sampling from an unconditional model provides a way to verify if the model’s overall distribution has changed.
-  I believe that the task this paper aims to address is important, and it provides valuable insights through some new tests.

**Weaknesses:**

- Although this is a paper focused on benchmarking and analysis, the number of concepts used was limited, and there were no experiments related to violence, nudity, or copyright issues - topics of particular interest in unlearning. Wouldn’t it be more beneficial to increase the number of concepts rather than reduce the number of prompts for each?

- The value of some additional performance analyses is unclear for me.
  - In Figure 2, using simple and diverse prompts does not seem to add significant distinction. Doesn’t MS-COCO already contain a sufficient variety of simple and diverse prompts unlike the simple prompts for testing target concept?
  - It’s unclear why the task of selectively erasing target concepts in Section 4.3 is necessary. I believe a well-functioning unlearning algorithm may erase any prompt containing the target concept in real world scenario.
  - I think making the model to prevent generation even when the user provides highly specific information in the downstream task(e.g. HED inputs) can harm the model's performance on downstream tasks. The task is contradictory. (whether to follow user instruction or ignore to erase the concept)

- The paper would benefit from testing if the techniques remain robust against recently popular red-teaming methods, such as ring-a-bell and UnlearnDiffAtk.

**Questions:**

The questions are somewhat related to the disadvantages I wrote.
- Is it possible to add NSFW or copyrighted character for benchmark concept? Actually, the small number of the concept is the primary concern of this review as a benchmark paper.
- Could you explain the advantages of some of the additional analyses mentioned under weaknesses in more detail?

---

### Official Review · Reviewer_Y8xa · 2024-11-04

**Soundness:** 2
**Presentation:** 3
**Contribution:** 2
**Rating:** 5
**Confidence:** 3

**Summary:**

This paper introduces a holistic unlearning benchmark for text-to-image diffusion models, aimed at evaluating methods for removing undesired or potentially harmful content from generative models. The authors apply their benchmark to six existing unlearning methods, testing on four specific target concepts. The experimental results reveal that all tested methods have certain limitations or side effects when applied in practice.

**Strengths:**

- The paper is well-structured and clearly written, making it accessible and easy to follow.
- By proposing a new benchmark for unlearning, the paper addresses an emerging direction in generative model research. Given the increasing power and prevalence of generative models, it is timely and important to examine ways to prevent harmful content generation.

**Weaknesses:**

### Limited Scope and Specificity of the Proposed Benchmark

- **Narrow Methodology**: The benchmark focuses on only six existing unlearning methods, limiting its generalizability. It is tailored specifically to these methods rather than serving as a more widely applicable benchmark.
- **Restricted Target Concepts**: The benchmark is tested on just four target concepts, which may not sufficiently represent real-world applications. Additionally, unlearning is often applied to remove harmful or inappropriate content, yet the chosen concepts do not necessarily reflect this priority. Given this specificity, the generalization claims made in the limitations section may be overstated.
- **Dependency on GPT4o**: The evaluation heavily relies on GPT4o, introducing uncertainty into the results. As the GPT series rapidly evolves, the outcomes of the benchmark may vary across versions. While GPT4o capabilities may justify its use, further quantitative evidence is necessary to substantiate its role in the benchmark.

### Lack of Experimental and Theoretical Support for Key Takeaway Messages

The paper presents several takeaway messages, but many lack sufficient experimental evidence or theoretical grounding, limiting their generalizability. Key examples include:

- **Page 6**: The authors emphasize "diverse and complex prompts," yet the prompts used in the experiments are limited to a few specific concepts. Furthermore, the discussion on balancing faithfulness and prompt compliance does not introduce significant new insights, as similar evaluations exist in works like ImageReward and PickScore.
- **Page 7**: The concept of over-erasing is discussed, but it is primarily explored through specific, hand-crafted related concepts. For example, Appendix E categorizes a gas pump as a "Machine" and connects it to a coffee machine, while an English Springer is linked to other dog breeds. The considerable gap between these examples suggests that defining over-erasing and similar relationships requires a more comprehensive analysis.
- **Page 9**: Distributional differences are studied using MNIST, a dataset with limited relevance to more complex image datasets. Testing on larger, more diverse datasets, such as ImageNet, would strengthen this analysis.
- **Page 10**: The choice of downstream tasks and experimental setup may not align well with the study's primary focus on unlearning text-based concepts. The downstream tasks are primarily image-focused, creating a possible mismatch with the unlearning of textual content.

### Conclusion

The limited scope of methods and target concepts, coupled with the reliance on GPT4o, raises concerns about the generalizability and long-term relevance of this benchmark in a rapidly evolving field. The takeaway messages presented lack substantial new insights, with many findings aligning with existing knowledge or lacking sufficient experimental support. Furthermore, the paper does not adequately demonstrate how this benchmark could be applied to effectively prevent harmful content generation in real-world settings. For these reasons, the contribution of this work may not yet meet the level of significance required for publication in this venue.

**Questions:**

please see weakness points above.

---

> ### Comment · Reviewer_Y8xa · 2024-11-27
> **Final comments**
>
> Given that the authors have not offered a rebuttal and other reviewers also give negative ratings, I will remain my original score.

---

### Official Review · Reviewer_snv7 · 2024-11-09

**Soundness:** 3
**Presentation:** 3
**Contribution:** 3
**Rating:** 5
**Confidence:** 3

**Summary:**

The Holistic Unlearning Benchmark provides a comprehensive evaluation framework for text-to-image diffusion model unlearning. The benchmark assesses unlearning methods across five key aspects, revealing effectiveness on target concepts, faithfulness of images, compliance with prompts, robustness on side effects, and consistency in downstream applications. The authors aim to inspire further research for more reliable and effective unlearning methods by releasing the evaluation framework.

**Strengths:**

The paper introduces a novel and comprehensive framework specifically designed for benchmarking unlearning methods in text-to-image diffusion models, which is crucial for ensuring ethical and responsible AI usage. Specifically, the paper evaluates six methods from five different aspects. The experimental settings are clearly presented and the takeaway notes are interesting. Overall, it raises an awareness of evaluating unlearning methods for the generalization ability.

**Weaknesses:**

While the paper proposes an interesting evaluation framework for unlearning methods, it does not present an in-depth discussion in each experiment that technically analyzes how certain method affects the performance. Therefore, it lacks depth in providing a comprehensive technical analysis for current unlearning methods.

Unmatched Content to Purpose: For instance，in Sec.5.2, the mentioned purpose is to find out how does unlearning change the underlying estimated distribution, while the experiment and the discussion only answer which methods change the distribution. This makes the content somewhat unmatched to its subtitle.

**Questions:**

See weakness.

---

### Note · Authors · 2024-11-28

**Comment:**

We first thank all reviewers for their thoughtful and detailed feedback. We agree with their comments and appreciate valuable insights. After careful consideration, we have decided to withdraw our submission. In our future updates, we plan to address the feedback by expanding the scope of target concepts to include NSFW and copyright contents and providing a deeper analysis of the unlearning effects. Again, we sincerely appreciate the time and effort spent reviewing our work, and we look forward to submitting an improved version in the future.

**Withdrawal Confirmation:**

I have read and agree with the venue's withdrawal policy on behalf of myself and my co-authors.